# Developmental Pharmacokinetics of Antibiotics Used in Neonatal ICU: Focus on Preterm Infants

**DOI:** 10.3390/biomedicines11030940

**Published:** 2023-03-17

**Authors:** Olga I. Butranova, Elena A. Ushkalova, Sergey K. Zyryanov, Mikhail S. Chenkurov

**Affiliations:** 1Department of General and Clinical Pharmacology, Peoples’ Friendship University of Russia (RUDN University), Miklukho-Maklaya St. 6, 117198 Moscow, Russia; 2State Budgetary Institution of Healthcare of the City of Moscow “City Clinical Hospital No. 24 of the Moscow City Health Department”, Pistzovaya Srt. 10, 127015 Moscow, Russia

**Keywords:** preterm neonates, pharmacokinetics, absorption, distribution, metabolism, excretion, antibiotics, ontogeny

## Abstract

Neonatal Infections are among the most common reasons for admission to the intensive care unit. Neonatal sepsis (NS) significantly contributes to mortality rates. Empiric antibiotic therapy of NS recommended by current international guidelines includes benzylpenicillin, ampicillin/amoxicillin, and aminoglycosides (gentamicin). The rise of antibacterial resistance precipitates the growth of the use of antibiotics of the Watch (second, third, and fourth generations of cephalosporines, carbapenems, macrolides, glycopeptides, rifamycins, fluoroquinolones) and Reserve groups (fifth generation of cephalosporines, oxazolidinones, lipoglycopeptides, fosfomycin), which are associated with a less clinical experience and higher risks of toxic reactions. A proper dosing regimen is essential for effective and safe antibiotic therapy, but its choice in neonates is complicated with high variability in the maturation of organ systems affecting drug absorption, distribution, metabolism, and excretion. Changes in antibiotic pharmacokinetic parameters result in altered efficacy and safety. Population pharmacokinetics can help to prognosis outcomes of antibiotic therapy, but it should be considered that the neonatal population is heterogeneous, and this heterogeneity is mainly determined by gestational and postnatal age. Preterm neonates are common in clinical practice, and due to the different physiology compared to the full terms, constitute a specific neonatal subpopulation. The objective of this review is to summarize the evidence about the developmental changes (specific for preterm and full-term infants, separately) of pharmacokinetic parameters of antibiotics used in neonatal intensive care units.

## 1. Introduction

Bacterial infections are an important cause of mortality and morbidity in newborns especially in low and middle-income countries [1,2] hence antibiotics are among the most used drugs in this population. About a quarter of the top 100 drugs used in the neonatal intensive care unit (NICU) are antibiotics [3,4]. The most common reason for antibiotic therapy in NICU is neonatal sepsis (NS) [5]. In the cross-sectional study of late-preterm and full-term neonates (*n*—757,979, 11 countries of Europe, North America, and Australia) it was shown that the antibiotic exposure was 135 days per 1000 live births. The incidence of early-onset sepsis (EOS) was 0.49 cases per 1000 live births (range, 0.18–1.45 cases per 1000 live births), and the EOS-associated mortality rate was 3.20% [6]. A systematic review and meta-analysis of 26 publications (2,797,879 live births and 29,608 sepsis cases in 14 countries) revealed NS incidence equal to 2824 (95% credit interval, CI 1892 to 4194) cases per 100,000 live births with mortality of 17.6% (95% CI 10.3% to 28.6%) [7]. Estimation of global sepsis incidence using data from the Global Burden of Disease Study (2017) revealed a peak in early childhood with 2.9 million (95% uncertainty interval, UI 2.6–3.2) sepsis-related deaths among children less than 5 years old [8]. A systematic review and meta-analysis from data of 30 years (period from 1990 to 2019, 29 studies, 164,750 neonates with sepsis) highlighted the etiology of NS. Main causative agents included coagulase-negative staphylococci (*CoNS*)—33% (95% CI 24–43), *Escherichia coli*—17% (95% CI 13–20), and *Klebsiella* spp.—14% (95% CI 11–17) [9]. The etiology of NS in developing regions was studied in the systematic review and meta-analysis (included 152,217 infants): the main pathogens were *Klebsiella* spp. (26.36%), *Staphylococcus aureus* (23.22%), *CoNS* (23.22%), and *Escherichia coli* (15.30%) [10]. Studies, based on data from the Burden of Antibiotic Resistance in Neonates from Developing Societies (BARNARDS) network also proved the leading role of *Klebsiella* spp., especially *Klebsiella pneumoniae* in NS [11,12,13]. In very preterm infants with EOS, in the framework of the BARNERDS network, the main pathogen was *E. coli* [14].

Empirical antibiotics recommended for the management of NS by different guidelines include benzylpenicillin, ampicillin or amoxicillin, and aminoglycosides (mainly gentamicin) [15,16,17]. Pediatric survey data from 56 countries support that gentamicin and ampicillin are the most prescribed antibiotics in hospitalized neonates in Africa, the Americas, the Eastern Mediterranean, Europe, and South-East Asia, while in the Western Pacific region most common are amoxicillin in combination with β-lactamase inhibitor, ceftizoxime, and meropenem [18]. The global survey of antimicrobial use in NICUs revealed the leading prescriptions: ampicillin (40%), gentamicin (35%), amikacin (19%), vancomycin (15%), and meropenem (9%) for “rule-out” sepsis and “culture-negative” sepsis. Vancomycin (26%), amikacin (20%), and meropenem (16%) were the most prescribed agents in the definitive treatment of presumed/confirmed infection [5]. Depending on bacterial susceptibility profile and involvement of different organs and systems treatment of NS may also include linezolid, dalbavancin, ceftaroline fosamil, and ceftolozane tazobactam [19].

The rise of antibiotic resistance mediates the involvement of additional antibiotics in NS management. The BARNARDS observational cohort study revealed 390 Gram-negative isolates of which 379 (97.2%) were resistant to ampicillin and 274 (70.3%)—to gentamicin [12]. Among causative agents of EOS in very preterm infants 60.6% of *E. coli* isolates and 42.9% of *Klebsiella* spp. were multidrug resistant [14]. Increased rates of antibiotic resistance lead to the use of antibiotics by Watch and Reserve groups (World Health Organization classification), which may result in decreased safety profile. The study of antibiotics prescription data in 5267 neonates from 56 countries defined that the use of Watch antibiotics reached 64.5% in China, and maximum rates of Reserve antibiotics use were seen in Russia (13.9%) and Mexico (12.8%) [18]. There was an attempt in the review by Darlow CA et al. (2021) to propose additional five antibiotics to be used in novel empiric regimens for NS [20]. Among these antibiotics, only one belongs to the Access group—amikacin, while four—to the Watch group (tobramycin, flomoxef, and cefepime) and one to the Reserve group (fosfomycin) [21].

A high rate of mortality in NS is associated with a relatively high level of treatment failure: in the review of 90 randomized controlled trials (RCT), it was reported in 15 studies [22]. The current list of antibiotics used in NICU is broad and reasons for NS treatment failure may be drug-related, with administration and/or dosing errors [23,24,25,26], which can be attributed to poor physicians’ awareness about antibiotics pharmacokinetics and, thus, pharmacodynamics in full-term and especially premature newborns. Extrapolating adult antibiotic regimens to the neonatal population may result in increased toxicity or altered efficacy of treatment due to different degrees of organ and system maturation. The problem is compounded by the high heterogeneity of the neonatal population and the dependence of pharmacokinetic properties on postnatal age, gestational age, and postconceptional age. There is a lack of RCTs including neonates making the choice of a proper antibiotic regimen challenging, especially for preterm infants. Optimal antibiotic treatment of NS is proposed to be based on the next key components: correct treatment phase, correct antibiotic dose, proper antibiotic exposure, and microbiological response determined by minimum inhibitory concentration (MIC) of antibiotic and by the type of drug action, time- or concentration-dependent [27]. Achievement of a proper value of MIC, and/or needed time of exposure to antibiotics is based on the pharmacokinetic (PK) parameters of a drug which are highly variable in the neonatal population.

This review highlights the developmental changes in preterm and full-term neonates and their impact on PK parameters of antibiotics of various groups used in NICU practice.

## 2. General Considerations on Neonate’s Pharmacokinetics

Different rates of maturation of organs and systems in neonates, the different capacities of enzymatic systems together with pathological changes due to critical illness, inflammatory status, and therapeutic interventions underlie variability of PK parameters and, thus, variability of clinical response in the treatment of infectious diseases. Pharmacokinetics is directly connected with pharmacodynamics through the parameter of plasma drug concentration. Considering antibiotics, the value of plasma concentration is especially important and the next pharmacokinetic/pharmacodynamic (PK/PD) indices are used to describe antibiotic efficacy: ratio of the area under the concentration-time curve (*AUC*) from zero to 24 h (*AUC_0–24_*) to the minimum inhibitory concentration (*MIC*), the ratio of the maximum plasma concentration (*C_max_*) to the *MIC* and % of the time during which the free plasma concentration exceeds the *MIC* (*% fT > MIC*). Antibiotic concentration is also an important factor in the prevention of antibacterial resistance (Figure 1). Next parameters are considered from this point: the mutant prevention concentration (*MPC*), defined as the *MIC* of the least-susceptible, single-step mutant [28], the ratio of minimum concentration to *MIC* (*C_min_*/*MIC*), the ratio of *C_max_* to *MIC* (*C_max_*/*MIC*), the ratio of *AUC_0–24_* to *MPC* (*AUC_0–24_*/*MPC*) or to *MIC* (*AUC_0–24_*/*MIC*). In the systematic review by Sumi CD et al. (2019) values of the listed above ratios suppressing the emergence of antibiotic resistance were established for different groups of antibacterials. For β-lactam antibiotics, *C_min_*/*MIC* should be equal or exceed 4; for aminoglycosides, *C_max_/MIC* should be equal or exceed 20; for fluoroquinolones, *AUC_0–24_/MPC* should be ≥35; for tetracyclines, *AUC_0–24_*/*MIC* should be ≥50; for polymyxin, B *AUC_0–24_*/*MIC* should be ≥808; for fosfomycin, *AUC_0–24_*/*MIC* should be ≥3136 [29]. The value of plasma antibiotic concentration changes through the main pharmacokinetic phases: absorption, distribution, metabolism, and excretion. Figure 1 represents PK/PD targets for optimizing efficacy and minimizing antibacterial resistance for time-dependent, concentration-dependent, and concentration-dependent with time-dependent antibiotics.

### 2.1. Absorption and Bioavailability

Drug absorption is the process of drug transportation from the site of administration to the systemic circulation and the fraction of unmetabolized drug that reaches the systemic blood flow is bioavailability. There are two main measures of the extent and rate of drug absorption—*AUC* and *C_max_*. A systematic review and meta-analysis of 31 studies including neonates revealed decreased bioavailability of oral antibiotics with the delayed achievement of *C_max_* [30].

Absorption and bioavailability both depend on drug-related factors (formulation, rate of solubility, physicochemical properties, etc.) and patient-related factors. Patient-dependent factors affecting absorption and thus bioavailability in the oral route of drug administration include the value of pH in the stomach and intestine, the rate of motility of the gastrointestinal tract (GIT), GIT maturation rate, gastric emptying time, expression of transport proteins in GIT, the quality and quantity of gastric juices, the rate of bile production, pancreatic function, first-pass metabolism, gastric volume, and mucin production. The main sites of absorption of orally administered drugs in neonates include the stomach, small intestine, and colon. Absorption in the stomach is governed first by gastric pH and gastric emptying time. It is considered, that in neonates (both term and preterm) the colon can play a significant role in the absorption of some drugs and nutrients, which is not true for adults [31].

Neonates are characterized by different degrees of the structural and functional maturation of the GIT depending on the term or preterm birth mainly. The anatomical differentiation of the intestine happens within 20 weeks of gestation, while functional maturation demands more time and appears in general after 32–34 weeks of gestation [32]. Preterm newborns are characterized by a reduced intestinal surface area due to limited villi maturation (full maturation of villi is observed around 20 weeks of gestation), which may lead to impaired absorption of orally administered drugs compared to full-term neonates [33].

Gastric acid secretion emerges in the second trimester of pregnancy and is less in preterm babies compared with full-term; from the moment of birth in two months there is a doubling of the value of secretion [33]. Pepsinogen secretion starts after 17–18 weeks of gestation, and enterokinase—after 24 weeks (25% of the level of older infants). Lactase activity is decreased in preterm newborns, while activities of sucrase, maltase, and isomaltase have no differences with full-terms. Pancreatic lipase is decreased in preterm neonates, as well as bile production and bile ileal reabsorption. Concentrations of bile acids and bile salts in the intestinal lumen at birth are relatively low, and insufficient functioning of transporter-mediated uptake and enterohepatic bile circulation is observed, though passive bile reuptake in the intestine and active transport in the distal ileum are presented [31].

An important factor in drug solubility and thus, absorption, is gastric volume. According to the Biopharmaceutics classification system (BCS), drugs can change their solubility class at different ages with the change in gastric volume. For example, antibacterials including chloramphenicol, erythromycin, and cefalexin alter from a low solubility classification in 6-month-old children to high solubility in adults [34]. Gastric motility is low in preterm newborns of 28–32 gestational weeks, while after 32 weeks, the value is close to full-terms. Gastric emptying is considered to be slower in infants than in older children and adults, with discussible factors affecting certain values [35]. A meta-analysis including 49 published studies of 1457 individuals proved the absence of effects of postnatal age or gestational age on the mean gastric emptying time. A significant effect of meal type was established. The fastest emptying time (mean simulated gastric residence time of 45 min) was determined for aqueous solutions, and the slowest (98 min) for solid food [36]. Published data state that the time of intestinal transit in preterm newborns is four times longer than in adults [37].

The value of gastric pH in neonates, both full-term and preterm, is still a discussible question. Some works stated a nearly neutral pH at birth (6–8) with further change to acidic values [38]. In a study of 40 preterm infants (24–33 weeks of gestation) median gastric pH was found to be between 4.5 and 5.5 [39]. A study of 29 preterm neonates ≤28 weeks of gestational age with body weight ≤ 1000 g showed an acidic value of gastric pH on day 1 with nearly no change up to 4 weeks. For neonates ≥26 weeks, mean pH ± standard deviation on the first day was 5.25 ± 3.00, and on the 28th day, it was 4.56 ± 1.34. For neonates <26 weeks, the values were 5.33 ± 2.17 and 4.87 ± 1.06, correspondingly [40]. Research conducted in the earlier period with a smaller number of neonates revealed variable results with mean pH ranging from 1.3 up to 6.9 [41,42]. Postprandial pH in the infant stomach was shown to be on a level above 4.5 for about two hours [35]. In the intestinal lumen of infants, the range of pH was reported to vary between 5.8 and 7.0, based on limited data [31].

The rate of expression of transport proteins provides a great impact on drug absorption since it is generally low in neonates. There was proved a strong relationship between the age of neonate (mainly postmenstrual age) and expression of such transport proteins, as breast cancer resistant protein (BCRP), bile salt efflux pump (BSEP), glucose transporter 1 (GLUT1), P-glycoprotein (P-gp), multidrug-resistance like protein 1 (MRP1), 2 (MRP2), and 3 (MRP3), Na^+^-taurocholate cotransporting polypeptide (NTCP), organic anion transporter polypeptides 1B1 (OATP1B1), and organic cation transporter 1 (OCT1) [43]. In preterm newborns, it is important to consider a high probability of insufficient levels of some transport proteins administering drugs orally. In the intestine, efflux transporters limiting drug absorption are P-gp, MRP2, and BCRP. Drug intestinal absorption is mainly performed with OATP1A2, OATP2B1, and peptide transporter 1 (PEPT1) [44].

Characteristics of the main transport proteins affecting drug absorption in the intestine are shown in Table 1.

The oral route of drug administration is preferred in children including neonates whenever it is possible since this route is non-invasive and painless. The summary of neonatal physiological factors affecting the absorption and bioavailability of orally administered drugs is given in Table 2.

PK parameters of orally administered antibiotics in neonates were studied mainly for beta-lactams, and for many of them, plasma concentration values high enough to exceed MIC were demonstrated.

#### 2.1.1. Absorption of Oral Penicillins

Ampicillin is the most prescribed antibiotic used in NICU [4,5]. Considering other penicillins in NICU, in the work by Hsieh EM et al. (2014) five drugs were reported (amoxicillin, oxacillin, nafcillin, penicillin G, and piperacillin tazobactam), in the work by Prusakov P et al. (2019)—nine (amoxicillin, and amoxicillin clavulanate, ampicillin sulbactam, penicillin G, cloxacillin, oxacillin, nafcillin, piperacillin, and piperacillin tazobactam) [4,5].

From the middle of XX century PK parameters of many penicillins were studied in neonates, and the results revealed differences between oral and parenteral routs of administration and age-dependent changes [30]. Oral penicillin was characterized by a lower *C_max_* value compared with penicillin administered intramuscularly [68]. Studies revealed a better absorption of oral flucloxacillin in infants 0–1 month old compared with older children [69], especially for such formulations as mixtures [70], plasma concentrations after both intravenous (i.v.) and oral administration were well above the *MIC*-values generally reported for *Staphylococcus aureus* [71]. A review of the works dedicated to oral flucloxacillin in neonates revealed a *T_max_* of 2 h in a majority of studies [30]. Studies of oral ampicillin and amoxicillin in neonates revealed a delay of *T_max_* up to 4 h compared with 30 min in intramuscular administration with bioavailability values close to adults, *C_max_* in preterm neonates was the highest compared with full-terms and older children [30,72].

Amoxicillin clavulanate is one of the most prescribed oral antibiotics for both adult and pediatric populations used in case of proved or strictly suspected beta-lactamase production. Evaluation of PK parameters of oral clavulanic acid in term neonates (*n*—15, dose of amoxicillin clavulanate 25/6.25 mg/kg, thrice a day administration) resulted in great variance in plasma concentrations (median: 1.4 mg/L; range: 0.20–4.82 mg/L) and extrapolation led to *AUC* of at least 8.4 mg·h/L, which is comparable to those of adults [73]. The conclusion of high variability of PK parameters of clavulanic acid in the pediatric population is supported by a systemic literature review of 18 studies, which did also state that *C_max_*, for oral administration, was below that of the intravenous route [73], which is like it is in adults [74].

PK studies of penicillins in neonates with sepsis are limited. The results of the PK study of oral amoxicillin used together with parenteral gentamicin in 60 0–2 month infants with sepsis (final analysis included blood samples from 44 infants) revealed that amoxicillin concentrations exceed the susceptibility breakpoint for amoxicillin (2.0 mg/L) against resistant *S. pneumoniae* strains for >50% of a 12-h dosing interval [75].

#### 2.1.2. Absorption of Oral Cephalosporines

In the list of most used drugs in neonatal ICU (for the period 2005–2010 years) among cephalosporins there are seven positions, and oral formulation is available only for the one—for cephalexin [4]. In the global point-prevalence survey of antimicrobial use in neonatal ICUs, there were eight cephalosporins with two available in oral form—cefadroxil and cephalexin [5].

The pharmacokinetics of oral cephalosporins are described in several studies including pediatric population. In children with osteoarticular infection, the achievement of optimal plasma exposure was demonstrated for oral cephalexin (median dose 40 mg/kg/dose every 8 h) [76]. Data on PK of oral cephalexin in newborns revealed that the serum levels achieved after a dosage of 15 mg/kg 8 hourly were lower than the average *MIC* for most of the Gram-negative organisms causing infections in neonates, while the levels achieved after 50 mg/kg 12-hourly were close to those in adults receiving 1 g. *T_max_* was at about 2 h, demonstrating slower absorption than in adults [77]. PK studies performed in 70 s of XX century with cefadroxil and cephradine included infants and children with no inclusion of neonates. The results reported a decrease in *C_max_* in the fed state, which can be true also for the neonatal population [78,79]. PK of cefaclor was studied in 10 newborns, a mean peak serum concentration of 7.7 μg/mL was achieved at 1 h after an oral dose of 7.5 mg/kg [80]. For plasma concentrations of cefaclor, the same effects of fed and fasting states were detected as those for cefadroxil and cephradine [81]. On the opposite side, for cefprozil, there were no food effects on PK [82]. PK of oral cefuroxime axetil was studied in the pediatric population (minimum age of participants—3 months). This study reported achievement of concentrations exceeding the MICs for common respiratory tract pathogens, including beta-lactamase-producing strains of *Haemophilus influenzae* and *Moraxella catarrhalis*. Administration of 10 or 15 mg/kg doses resulted in serum cefuroxime concentrations similar to adult values following a 250 mg cefuroxime axetil tablet [83]. For oral cefixime pediatric data, supposed GIT absorption from 40 to 50% is more rapid and complete for oral suspension [84]. Oral ceftibuten (single oral dose of either 4.5 or 9.0 mg/kg) PK parameters were studied in infants and children. Rapid absorption was shown with mean *T_max_*—140 min and *C_max_* from 5.0 to 19.0 mg/L with no differences between dosing regimens [85]. Pediatric PK studies of cefpodoxime revealed *T_max_* prolongation in the fed state (fed = 2.79 ± 1.10 h vs. fasted = 1.93 ± 0.54 h), but the extent of absorption was not affected by food [86]. Though this result was derived from the pediatric population, it can be also considered actual for neonates.

#### 2.1.3. Absorption of Oral Carbapenems

The first carbapenem with the oral route of administration approved for pediatric practice in Japan is tebipenem. Though there are no data on neonates, it is interesting to consider PK parameters derived from pediatric studies (*n*—217, minimum age of participant—6 months, dosing regimen—twice a day at 4 mg/kg or 6 mg/kg for 8 days). *T_max_* values were 0.74 ± 0.26 and 0.69 ± 0.22 h (average + standard deviation) in 4 mg/kg and 6 mg/kg, *C_max_* values were 3.48 ± 1.67 and 5.20 ± 2.84 mg/mL, and the *AUC_0–24 h_* values were 11.00 ± 1.84 and 16.07 ± 3.35 mg·h/mL in the 4 and 6 mg/kg dosage groups, respectively [87]. Tebipenem pivoxil is characterized by remarkably high absorption in GIT due to its ability to be transported by OATP1A2 and OATP2B1 along with simple diffusion [63]. Interestingly to note, tebipenem pivoxil is the first beta-lactam demonstrated OATP-mediated transport in the process of intestinal absorption. Since there are some studies stating an increased intestinal OATP2B1 expression in neonates [55], there is a potential for variability in intestinal tebipenem pivoxil absorption in this population.

#### 2.1.4. Absorption of Oral Macrolides

Macrolides are antibiotics used mainly in ambulatory practice, but in the list of 100 drugs most used in NICU erythromycin was ranked 24th being the 6th among most used antibiotics and the only macrolide [4]. In the global survey by Prusakov P et al. (2021), there were three macrolides used in neonatal ICU, erythromycin, azithromycin, and clarithromycin [5]. Erythromycin was the first macrolide studied in children. Comparison of absorption of erythromycin suspension in three age groups (0–1 month, 1–6 months, and 6 months to 6 years) revealed the lowest value for infants less than 1 month of age [88]. The value of absolute bioavailability of rectally administered erythromycin was 28% in neonates, 36% in infants, and 54% in children greater than 1 year [89].

PK parameters of orally administered azithromycin were studied in children after a single oral dose of 10 mg per kg of body weight on day 1 followed by single daily doses of 5 mg/kg on days 2 to 5. Mean values of PK parameters (±standard deviation) were estimated for *C_max_*, *T_max_*, and *AUC_0–24_*: 383 + 142 ng/mL, 2.4 + 1.1 h, and 3109 + 1033 ng × h/mL, respectively. Concentrations in serum at 0 h (predose) and at 24, 48, and 72 h after the final dose were 67 + 31, 64 + 24, 41 + 17, and 29 + 14 ng/mL, respectively [90]. In the study of single (*n*—14) and multiple oral doses (*n*—9) of 12 mg/kg in 23 children, *C_max_* and *T_max_* of azithromycin were 318.2 + 174.5 μg/L, and 2.4 + 1.1 h, respectively, with no differences estimated between these doses groups [91]. Comparison between 30 mg/kg immediate-release (IR) and 60 mg/kg extended-release (ER) forms of azithromycin oral suspension in children revealed similar or greater systemic exposure in the case of ER form [92].

PK studies including clarithromycin revealed high rates of oral absorption both in children and adults not affected by food. In infants and children (6 months to 10 years), a brief delay in the onset of absorption was demonstrated for clarithromycin (suspension, 7.5 mg/kg). The mean *C_max_* for clarithromycin was reached within about 3 h both under fasting and fed conditions with corresponding values of 3.59 and 4.58 micrograms/mL in a single-dose regimen. *C_max_* values for 14-(R)-hydroxylated metabolite for fasting and fed conditions were 1.19 and 1.26 micrograms/mL, respectively [93]. Other studies reported a similar profiles of clarithromycin pharmacokinetics in the pediatric population [94].

#### 2.1.5. Absorption of Oral Oxazolidinones

Linezolid is used in neonates including preterm ones. Its bioavailability is extremely high in the oral route, reaching 100% with minimum decrease under the fed condition, *T_max_* is about 1–2 h [95]. The PK study of oral linezolid suspension, 10 mg/kg used in 4-month-old infants (birth at 25 weeks) revealed that *C_max_* and *AUC* were lower than in full-term infants who used linezolid intravenously [96]. Possible factors affecting the PK of linezolid in neonates may include incomplete absorption, faster clearance, or a smaller actual volume of distribution. PK study including extremely premature infants was performed for oral linezolid (dose 10 mg/kg every 8 h) and continuous intravenous infusion (dose 30 mg/kg). Analysis of 7 serum samples for oral linezolid revealed a mean *C_max_* equal to 9.04 (0.69–32.9) mg/L, and 17 serum samples for intravenous linezolid revealed a mean *C_max_* equal to 1723 (2.6–30.4) mg/L. Reported *C_max_* values for both routes of administration were ≥*MIC* for causative microflora [97].

Another oxazolidinone, tedizolide, is also used in pediatric practice. The study of oral and intravenous tedizolid in infants (age—1 day to 24 months) started in the 2017 year (NCT03217565) and still has a recruitment status [98]. PK parameters of oral tedizolide (dose 3–6 mg/kg) were studied in children (2 to <12 years old), and results revealed high bioavailability value, compared with that of the intravenous route, median *T_max_* was 2–3 h compared with 1–2 h after initiation of the 1 h intravenous infusion. *C_max_* values were higher compared with adults (4.19 vs. ~2.5 mg/L) [99]. A study of oral and intravenous tedizolid in adolescents revealed no significant differences in PK parameters and the received results demonstrated similarity with adult PK parameters of tedizolid [100].

#### 2.1.6. Absorption of Oral Fluoroquinolones

Fluoroquinolones are among the antibacterials prescribed for neonates with infections, including ICU practice [5]. Ciprofloxacin is used in the majority of clinical situations intravenously, though in some situations, oral route is available. Population pharmacokinetics of ciprofloxacin including oral form was derived using data from a mixed population, including 3 newborns, 17 infants and toddlers, 27 children, and 8 adolescents and young adults. For i.v. ciprofloxacin, sampling at a single point (12 h after the start of infusion) allowed the precise and accurate estimation of clearance (CL) and the elimination half-life, as well as the ciprofloxacin concentration at the end of the infusion, for oral ciprofloxacin the presence of a lag time after administration suggests a schedule based on two sampling times of 1 and 12 h [101]. Another population PK study was performed with data from 60 newborn infants who used intravenous ciprofloxacin (postmenstrual age [PMA] range, 24.9 to 47.9 weeks). The main covariates affecting ciprofloxacin pharmacokinetics were gestational age, postnatal age, current weight, serum creatinine concentration, and use of inotropes. Monte Carlo simulation demonstrated that 90% of hypothetical newborns with a PMA of <34 weeks treated with 7.5 mg/kg twice daily and 84% of newborns with a PMA ≥34 weeks and young infants receiving 12.5 mg/kg twice daily would reach the *AUC/MIC* target of 125, using the standard EUCAST *MIC* susceptibility breakpoint of 0.5 mg/L [102].

Levofloxacin also may be administered orally in pediatric practice. In the study 85 children (6 months to <2 years, 2 to <5 years, 5 to <10 years, 10 to <12 years, and 12 to 16 years) received a single 7 mg/kg dose of levofloxacin (intravenously or orally) absorption demonstrated no age-dependence and was close to adult values [103].

The rate of oral absorption of other fluoroquinolones, moxifloxacin, and gatifloxacin, were not studied in infants and only extrapolation of the results of PK studies made for an older category of patients is possible. For oral moxifloxacin data are available for the pediatric population suffering from multidrug-resistant tuberculosis (*n*—23, the median age—11.1 years, 6 out of 23 were human immunodeficiency virus (HIV)-infected). The median *C_max_* was 3.08 (IQR, 2.85–3.82) µg/mL, *AUC* from 0–8 h (*AUC_0–8_*)—17.24 (interquartile range, IQR, 14.47–21.99) µg × h/mL, *T_max_*—2.0 (IQR, 1.0–8.0) h. In HIV-infected children, there was a decrease in *AUC_0–8_* and *C_max_. T_max_* was shorter with crushed vs. whole tablets [104]. PK parameters of oral gatifloxacin were studied for a single dose of suspension (doses—5, 10, or 15 mg/kg of body weight, 600 mg maximum; 76 patients with average age 6.7 ± 5.0 years) and for tablets (dose 10 mg/kg, 2 children >6 years of age). *C_max_* and *AUC* increased in a manner approximately proportional to the dose, and at the 10 mg/kg dose, the bioavailability was similar between the suspension and tablet formulation [105].

#### 2.1.7. Absorption of Other Oral Antibacterial Agents

Vancomycin is a glycopeptide administered intravenously to treat neonatal sepsis caused by resistant Gram-positive microflora. Oral vancomycin demonstrated effectiveness in the prophylaxis of necrotizing enterocolitis in preterm, very low birthweight infants, and no serious adverse effects were detected, with negligible serum drug concentrations suggesting the absence of systemic absorption [106]. The same results were revealed for older patients with colitis caused by *Clostridium dificile* (*n*—8, age from 2 to 18 years)—serum vancomycin levels were undetectable [107]. Published works reported increased rates of oral absorption of vancomycin in pediatric and adult patients with cancer and associated chemotherapy [108,109].

Trimethoprim sulfamethoxazole (TMPX) being a synthetic antimicrobial agent is found among drugs used in NICU in infants ≥3 days old [5]. Simulation of TMPX exposure (oral route of administration) revealed that for all dosing regimens steady-state area under the concentration-versus-time curve from time zero to τ (*AUC_0–τ,ss_*; where τ denotes the dosing interval) in subjects 0 to <2 years was similar with the group of 2 to <6 years of age (within 20%), but 29% less than in older children (6 to <21 years of age). Values of simulated *AUC_0–τ,ss_* both for infants and children were generally lower than that for 70-kg adults [110].

Clindamycin is used in neonatal ICU including infants <3 days old [5]. Clindamycin exists in oral form, though due to negative organoleptic properties, it is not a common choice for the pediatric population. Since clindamycin is a highly lipophilic agent, its absorption is thought to be unaffected by age-associated changes specific to the pediatric population including neonates [111].

Fosfomycin (C3H7O4P) is a phosphonic acid derivative representing an epoxide class of antibiotics [112] proposed to treat NS in resistance to first-line antibiotics [113]. Fosomycin is hydrolyzed in the stomach and absorbed in the small intestine; factors affecting its bioavailability include gastric pH and gastric emptying rate [114]. In the PK study of fosfomycin in neonates with suspected sepsis (The NeoFosfo study (NCT03453177)) oral bioavailability was estimated to be 0.48 (dose was 100 mg/kg) [114] that is close to adult values [114].

Rifampin is a macrocyclic antibiotic effective mainly against resistant Staphylococcus aureus and Mycobacterium tuberculosis. Oral forms may contain only rifampicin, or its combination of isoniazid and pyrazinamide. The study included children with tuberculosis (age 3 months to 13 years, dosing for initial treatment, intensive phase—rifampicin 60 mg, isoniazid 30 mg, and pyrazinamide 150 mg; for continuation phase—rifampicin 60 mg and isoniazid 30 mg) the mean *AUC_0–6_* on enrolment was 14.88 and 18.07 μg/hour/mL (*p* = 0.25) in HIV-infected and HIV-uninfected children, respectively, and after 4 months of treatment 16.52 and 17.94 μg/hour/mL (*p* = 0.59). The *AUC_0–6_* of all 55 children was 16.81 (+10.82) μg/hour/mL on enrolment and 17.39 (+9.74) μg/hour/mL after 4 months of treatment [115]. Previous PK studies of oral rifampicin in children revealed Cmax three times less than that for the intravenous route [116]. Using PK data from preterm and term infants dosing simulation was performed based on weight and postnatal age, simulated regimens resulted in comparable exposures to adults receiving therapeutic doses of rifampin against staphylococcal infections and TB [117].

#### 2.1.8. Effect of Food on Antibiotics Absorption

An important factor that may alter the rate of absorption of oral drugs is the presence of food in GIT. PK studies described above provide some data in this respect, and Table 3 represents available information on food effects on the absorption of oral antibiotics.

#### 2.1.9. Absorption in Non-Oral Routes of Administration

The data on age-associated change of absorption for non-oral routes of drug administration are given in Table 4.

### 2.2. Distribution

The process of drug distribution to various organs and systems is essential for the further interaction of drug molecules with their targets, demonstrating a link to pharmacodynamics. Distribution depends both on drug-associated factors and patient-associated factors. Physicochemical properties of drugs, such as lipophilicity and hydrophilicity, can dramatically affect passage through biological membranes since there is a direct relation between permeability and lipophilicity. Hydrophilic antibiotics include beta-lactams, aminoglycosides, glycopeptides, oxazolidinones (tedizolid), and antibacterials of different groups, such as colistimethate sodium, dalbavancin, fosfomycin, telavancin. General features of hydrophilic antibiotics: small size of molecules, relatively low molecular weight, distribution into the extracellular fluid, and low values of the volume of distribution (*Vd*), reflecting their propensity to remain in the plasma. Lipophilic antibacterials are fluoroquinolones, macrolides and ketolides, metronidazole, rifampicin, trimethoprim/sulfamethoxazole, linezolid (moderate lipophilic). For these agents, high *Vd* values are specific because of their ability to cross biological membranes with passive diffusion and reach extravascular spaces. Distribution of the same drug may vary depending on health state: in critically ill patients hydrophilic antibiotics (β-lactams, aminoglycosides, glycopeptides) tend to have an increased *Vd* and a reduced clearance, PK parameters of those, which are lipophilic, are less affected. 

Table 5 includes data on the distribution of different groups of antibiotics.

Patient-associated factors affecting *Vd* include water and fat content in the organism, proportion of extracellular fluids and total water, rate of functioning of the cardiovascular system, the quantity of plasma proteins, structure and function of blood-tissue barriers. Through the first year of life, and organism undergoes significant changes. There is an increase in weight by 50% in the first 6 weeks of life, doubling in the first 3–4 months and tripling at the end of the year. Through the infant period body length increases by 50% and body surface area doubles [123].

Total body water in neonates is higher than in older children and adults. Increased water proportion in the organism of neonate results in increased values of *Vd* of hydrophilic drugs and thus, leads to the need to use higher doses per kg [124]. Recent systematic review and meta-analysis (22 articles included in the final analysis) revealed that the mean total body water for full-term newborns was 73.8% (95% CI 72.47% to 75.06%, 15 studies, 433 infants). Meta-regression demonstrated that it was higher in preterm infants (up to 90% at 26 weeks gestation, dropping to 75% at 36 weeks corrected gestation) and had a negative correlation with gestation at birth, falling 1.44% per week (95% CI 0.63% to 2.24%, 9 studies, 179 infants) [125]. INTERGROWTH-21st Project was aimed at the estimation of composition patterns of the body of neonates (1019 newborns). The mean body fat percentage (±standard deviation) at birth was 9.6 ± 4.0 for boys and 10.7 ± 4.0 for girls. The fat mass demonstrated a moderate increase with gestational age with very large variability, 36 g/week (95% confidence interval (CI) 28–43 g) for girls and 33 g/week (95% CI 25–42 g) for boys. The mean value of fat-free mass (±standard deviation) was 2965 ± 422 g for boys and 2739 ± 320 g for girls [126]. Investigation of the association between preterm birth and body composition at 6, 18, and 30 years of age suggested that in boys preterm birth was associated with decreased body fat and fat-free mass in early age with subsequent increase in fat mass in adult age [127]. Published data reported a smaller percentage of fat mass in preterm newborns (3% in a 1.5 kg premature neonate) compared with full-terms (12%) [128].

The plasma protein binding (PPB) rate correlates with the amount of pharmacologically active, unbound fraction of a drug, which is responsible for the realization of the main drug action in the organism since it is the unbound fraction, which penetrates the extravascular space. Decreased amount of plasma proteins results in the increase in the unbound fraction and possible increase in toxicity. On the opposite side, an increased binding rate will result in the reduction of free, active drug concentrations, which may lead to the decreased antimicrobial activity of antibacterial agents in the site of infection, which is extravascular typically. It was also shown that the rate of plasma protein binding of antibiotics can affect the extent of their antimicrobial activity [129].

The main plasma proteins binding drugs are human serum albumin (HSA) and α-1-acid glycoprotein (AAG). The mean value of HSA for premature neonates was estimated at the level of 30.6 ± 4.7 g/L in the study including 199 subjects with a mean birth weight of 1272 ± 390 g, mean gestational age of 29.2 ± 2.2 weeks [130], less values were reported in another study: 25 ± 0.6 g/L in the population of 24 premature infants (mean gestational age 28.4 ± 0.4 weeks, mean birth weight 1080 ± 75 g) [131]. Reported values for HSA in neonates are less than in the adult population, resulting in a decrease in plasma protein binding rates. HSA is typically associated with binding acidic exogenous compounds.

AAG is an acute-phase reactant, and its levels are different in healthy subjects and in those with injury, inflammation, or infection. AAG displays a high affinity towards basic lipophilic compounds. In healthy subjects, plasma concentrations of AAG range from ≈50–130 mg/dL [132]. In neonates, AAG levels are less than those in adults. The ontogeny of plasma AAG in healthy subjects (average postnatal age from 0 days to 79 years) was studied using data from 26 separate studies. Maximum plasma AAG concentration (AAG_max_) in healthy adults was 93.17 mg/dL and the model estimates median AAG concentrations to be approximately 3.8-fold lower (i.e., 24.67 mg/dL) during the first day of life [133]. The ontogeny of AAG in subjects diagnosed or suspected of infection was studied using 214 individual AAG concentrations (postnatal age range: from 5 days to 20.5 years). For adults with suspected or diagnosed infection, AAG_max_ was 254.71 mg/dL, and increased AAG concentrations were specific for all ages: at 5 days old, median AAG concentrations were 89.41 mg/dL in infected individuals compared to 41.51 mg/dL in healthy subjects [133]. Factors affecting AAG levels also include gestational age and mode of delivery. AAG levels were found to be higher in infants born vaginally compared with those who were delivered by Caesarean section, and increased gestational age was associated with higher AAG concentrations [134].

An increase or decrease in the PPB rate may produce a significant change in the clinical response while using highly protein-bound drugs (≥90%). In the review by Meesters K et al. (2023), it was summarized that among penicillins relatively high PPB rates were reported for cloxacillin (≈94%), dicloxacillin (96–97%), flucloxacillin (95–96%), and oxacillin (92–96%) [135].

Among antibiotics with high rates of PPB, significant differences were demonstrated for different age groups and for the different health statuses of patients. For oxacillin, cloxacillin, and flucloxacillin an increase in the unbound fraction was demonstrated in hypoproteinemic patients resulting in overdose and risks of toxicity [136,137,138]. Neonatal plasma concentrations of HAS and AAG are less than in adults, resulting in a potential increase in the unbound fraction. In a study of 56 infants aged 3 to 87 days (gestational age, 25–41 weeks) flucloxacillin mean protein binding was 74.5% ± 13.1% with high variability (34.3% to 89.7%) [139].

The results of the systemic review of studies dedicated to PPB of different cephalosporines revealed a high rate of variability of binding patterns considering different fluids and different categories of patients [140]. The highest rates of protein binding in plasma and serum of adults were detected for cefonicid (mean 82–98%, four studies), and cefoperazone (mean 89–92%, four studies). The lowest protein binding rates were demonstrated for cefuroxime (mean 16–33%, five studies), cefotaxime (mean 8–41%, four studies), and ceftazidime (mean 0–21%, six studies) [140]. Estimating concentration ranges authors revealed a linear pattern of protein binding for the next cephalosporines: ceforanide, cefamandole, cefmenoxime, cefotaxime, cefluprenam, cefotetan, and cefoxitin. A typical non-linear pattern was found for cefazolin, cefonicid, cefpiramide, and ceftriaxone [140].

Several studies describe the PPB of cephalosporines in neonates and children. Age-dependent differences were detected in the PK study of cefoperazone. The unbound fraction of cefoperazone in newborns (*n*—17, mean age 35.7 (30.1–42.3) gestation weeks) was 24.6 (11.3–48.0)%, in infants (*n*—10, mean age 0.46 (0.1–2.0) years)—16.9 (9.1–27.4)%, in children (*n*—19, mean age 6.4 (2.2–9.0) years)—11.7 (8.1–18.6)% [141]. In healthy adults, ceftriaxone PPB was 95% [142]. Prediction of unbound ceftriaxone concentration in children resulted in high variability from 4.75% to 39.97% [143]. In critically ill children (0 to 18 years of age) median unbound fraction of ceftriaxone was 13.6 (7.6–70.3)% [144]. In critically ill adult patients unbound fraction of ceftriaxone was estimated at the level of 33.0 (20.2–44.5)% [145].

The highest rate of PPB among carbapenems is for ertapenem. Depending on drug concentration it varies from ~95% at concentrations of <50 μg/mL to ~92% at concentrations of 150 μg/mL (concentration at the end of a 30-min infusion following the 1-g dose) [146]. In critically ill patients, protein binding of ertapenem was also shown to be concentration-dependent and varied from 84% to 96% [147]. The unbound fraction of ertapenem in elderly subjects (~5 to 11%) was generally greater than that in young adults (~5 to 8%) [148].

High rates of PPB are also specific for glycopeptide teicoplanin and lipoglycopeptides. Teicoplanin has plasma protein binding of 90 to 95% in adults [149]. In neonates, this value was demonstrated to be lower—from 80.5% to 71.9% [150]. Though vancomycin has a moderate protein binding rate, PK studies revealed the same tendency: the unbound vancomycin fraction in neonates was higher than in children and adults, and total vancomycin concentration and albumin were the most important covariates of unbound vancomycin concentration [151]. The median fraction unbound of vancomycin was reported to be 0.9 in neonates, and 0.6 in adults [151].

Among lipoglycopeptides, the highest rate of PPB was reported for dalbavancin, 99% [152], for telavancin >90% [153], and for oritavancin ≈ 85% [154]. Dalbavancin clearance change in critically ill patients may propose the effect of hypoalbuminemia on PK parameters [155].

Some macrolides are characterized by relatively high PPB: for erythromycin, it varies from 80 to 93% [156], and for clarithromycin it is about 70% [157]. Azithromycin PPB is about 30% in adults, and the percent binding in newborns, especially preterm, is discussable [158], and for PK simulation in neonates, a PPB of 30% is used [159].

Among oxazolidinones, linezolid has low PPB: its unbound fraction was 85.4 ± 3.7% in plasma samples from adult surgical patients and 92.1 ± 6.2% in adult ICU patients [160]. In the pediatric population (*n*—15, 0–13 years old), median unbound linezolid concentration was 7.5 mg/L while the median total concentration was 9.4 mg/L, which results in a 79.8% unbound fraction [161]. PPB for tedizolid is from 70 to 90% [162]. It is thought that in critically ill patients, in a hypoalbuminemia state, the PPB of tedizolid will be changed resulting in increased clearance [163].

Lincosamides are characterized by variable values of PPB. Clindamycin has high rates of PPB, from 78 to 94%, and it binds mainly AAG. Decreased concentrations of AAG specific for neonates are expected to affect clindamycin’s distribution into tissues [164]. Another lincosamide, lincomycin, possesses concentration-dependent PPB, varying from 28 to 86% and decreasing with drug plasma concentration increase [165].

A lipopeptide antibiotic daptomycin has a high PPB and binds to several proteins, first to HSA and α-1-acid-glycoprotein [166]. In adults PPB is between 90–94%, it does not depend on concentration, but is decreased with decreasing renal function [167]. The clinical significance of this fact is discussible, in the study of physiologically based pharmacokinetic modeling of daptomycin dose optimization in pediatric patients with renal impairment no dose adjustment was proposed for patients with mild-to-moderate renal impairment, and only for severe cases was a dose decrease recommended [168].

Fluoroquinolone levofloxacin binds both to albumin and globulin, PPB is from 24 to 38% [169] and changes in PK parameters in the hypoalbuminemia state are not expected [170]. PPB for moxifloxacin is about 50% and does not depend on drug concentration [171]. No studies reporting PPB changes in term and preterm neonates are available for these drugs.

Fosfomycin has a negligible PPB rate and change in plasma protein concentration is not an important factor here [172]. Other antibiotics with very low PPB rates include aminoglycosides, so change in PPB is not expected in neonates [20]. 

Rifampin PPB rate is from 72 to 92% in adults with tuberculosis [173] and is not dependent on drug concentration in plasma [174]. In PK studies of neonates, no certain values of PPB were estimated for rifampin, though clearance increase was reported with postnatal age increase [117].

An important characteristic of antibiotics with high PPB rates is their ability to displace bilirubin from its binding with albumin, which may lead to bilirubin-induced neurotoxicity in neonates with hyperbilirubinemia. In Table 6, examples of antibacterials with different bilirubin displacing activity are given.

Bilirubin displacing activity of drugs, listed above, was mainly reported using data from in vitro studies. Sulfonamides are among the drugs commonly thought to be responsible for kernicterus in neonates. Currently, one of the most used sulfonamides is the combined drug, sulfamethoxazole-trimethoprim. For sulfamethoxazole, there is a 70% PPB [176], and for trimethoprim, PPB is about 44% [177]. The review of clinical and animal studies with cotrimoxazole revealed no strict evidence of kernicterus caused by sulfamethoxazole-trimethoprim in neonates. Authors made a conclusion, that oral administration to neonates for 7–10 days is unlikely to cause kernicterus [178].

Ceftriaxone is another drug considered to have a high potential to displace bilirubin. Clinical studies reported that in neonates (at least 15 days old) a short-term course of ceftriaxone did not have a higher likelihood of developing hyperbilirubinemia compared with a short-term course of cefotaxime [179]. Another recent study also demonstrated the absence of ceftriaxone association with a bilirubin-displacing effect in infants with a mild unconjugated hyperbilirubinemia (*n*—27, born at term, <7 days old, diagnosed sepsis) [180].

### 2.3. Metabolism

The primary site for drug metabolism is the liver; others are the kidney, intestine, lungs, and plasma. Biotransformation of drugs is realized in two phases with the assistance of drug-metabolizing enzymes (DMEs). DMEs of phase I are responsible for oxidation, reduction, and hydrolysis reactions. DMEs of phase II are responsible for glucuronidation, sulfation, methylation, and acetylation reactions.

Biotransformation of drugs may lead to: (1) inactivation (e.g., chloramphenicol), which is terminating drug action to excretion, and (2) active metabolite formation from a prodrug. Considering antibiotics prodrugs are available for several pharmacological groups. Among the penicillin group, there are several prodrugs of ampicillin aimed at the improvement of its bioavailability (pivampicillin, talampicillin, bacampicillin, and hetacillin), or the improvement of its pharmacodynamic properties (sultamicillin, a prodrug that links ampicillin and sulbactam by a methylene group) [181]. Highly effective cephalosporins in the form of prodrugs are ceftaroline fosamil [182] and ceftobiprole medocaril [183]. New carbapenem tebipenem became orally available in the form of a prodrug, tebipenem pivoxil [184]. Another prodrug among synthetic antibacterials is metronidazole [185].

The most important enzymes responsible for phase I reactions are cytochrome P450 (CYP450) enzymes with the most significant role of CYP3A4 isoenzyme, which is responsible for the biotransformation of many drugs including antibiotics (erythromycin, linezolid, clindamycin, etc.).

Some antibiotics are metabolized involving not only the most common isoforms but are also less frequent. For example, it was recently demonstrated, that linezolid metabolism is performed with the assistance of several isoenzymes in next order CYP2J2 ≫ CYP4F2 > CYP2C8 > CYP1B1 ≈ CYP2D6 ≈ CYP3A4 > CYP1A1 > CYP3A5 [186].

Developmental changes in the metabolic activity of CYP450 isoenzymes may result in changes in antibiotics pharmacodynamics. In general, expression of DME is typically low at birth and gradually increases with age [187,188]. The level of CYP-dependent metabolism at birth is proposed to be 50–70% of the adult value. The mean total CYP450 protein expression measured with western blot analysis was 390.3 pmol/mg in an infant-neonate group (2 fetuses, 2 newborns, 6 infants), and 644.59 pmol/mg in the >1 year group (6 children, 4 adults, age 2–72 years) [189,190]. Unlike children and adults with dominating CYP3A4, the main fetal hepatic CYP450 isoenzyme is CYP3A7, which accounts for up to 50% of the total fetal hepatic CYP content and up to 87–100% of total fetal hepatic CYP3A content [191].

Three different developmental patterns (classes) were defined for hepatic DME by Hines RN et al. (2013) [192]:First pattern (Class 1)—DME high in fetal life and low or absent after birth;Second pattern (Class 2)—DME stable throughout development;Third pattern (Class 3)—DME low in fetal life to increasing and high after birth.

Considering only CYP450, in the Class 1 there is only CYP3A7, in the Class 2—CYP2B6, CYP2C19, CYP3A5, and in the Class 3—CYP1A2, CYP2C9, CYP2D6, CYP2E1, CYP3A4 [192].

In the work by Fanni D et al. (2014), the activity of CYP1A2, CYP2C9, and CYP3A4 isoforms in 2 to 3 years old children was reported to be higher than in adults, while the activity of CYP2C19, CYP2D6 was similar to that in adults [187].

In Table 7 data on CYP450 isoenzyme ontogeny are given with information about antibiotics-substrates.

Drug interactions can significantly affect the treatment outcome. Drug interactions on the level of metabolism include interaction between CYP450 isoenzyme substrate and isoenzyme inhibitor or inducer, though substrate-substrate interactions are also possible. Among antibiotics, the most significant effect on CYP450 isoenzymes is produced by fourteen-membered ring macrolides (inhibition of CYP3A4 and 3A5) [205] and by rifampicin and its relatives (induction of CYP 3A4) [206].

Studies of fluoroquinolones revealed ciprofloxacin as a relatively strong inhibitor (≥5-fold increase in *AUC* or >80% decrease in clearance) of CYP1A2 [207]. Levofloxacin demonstrated inhibition of CYP2C9, and negligible inhibition of CYP1A2 and CYP2C9 was reported for caderofloxacin, antofloxacin, moxifloxacin, and gatifloxacin [208]. On the opposite side, novel fluoroquinolone compounds containing cyclopropylamine in the pyrrolidine ring of the 7′ position of fluoroquinolone molecule were demonstrated to be irreversible inhibitors of CYP3A [209,210].

Piperacillin was demonstrated to inhibit CYP2C8 [211], while other penicillins can induce isoenzymes. Flucloxacillin revealed a potential to induce expression of CYP3A4 and P-glycoprotein, most likely through the activation of the nuclear hormone receptor PXR [212], and dicloxacillin was shown to be an inducer of CYP2C19, CYP2C9, and CYP3A4 [53]. For carbapenem meropenem, there was demonstrated a substantial concentration-dependent interaction with voriconazole in a patient treated by these two drugs, and in vitro study confirmed the inhibitor action of meropenem on CYP3A4 and CYP2C19 [213].

Oritavancin produces a nonspecific weak inhibitory effect on CYP2C9 and CYP2C19, CYP1A2, CYP2B6, CYP2D6, CYP3A4, and simultaneously it is an inducer of CYP3A4 and CYP2D6 [214].

Antibiotics that can induce or inhibit main CYP450 isoenzymes are given in Table 8.

Other important DMEs of phase 1 are flavin-containing mono-oxygenase 3 (FMO3), hepatic carboxylesterases 1 (CES1) and -2 (CES2), and alcohol and aldehyde dehydrogenases (ADHs and ALDH, respectively) [189,194].

Studies revealed a lower expression of non-CYP I phase enzymes in neonates and pediatrics than in adults (ADH1A, ADH1B, ADH1C, ALDH1A1, CES1, CES2, and FMO3) [196]. CES1 abundance in hepatic microsomes revealed a 5-fold increase from neonates to adults (315.2 pmol/mg vs. 1664.4 pmol/mg), for CES2—a 3-fold increase (59.8 pmol/mg vs. 174.1 pmol/mg, respectively) [220].

FMO1 expression was reported to be the highest in the embryo (8–15 weeks of gestation; 7.8 ± 5.3 pmol/mg protein) with subsequent decrease, while FMO3 levels were low in embryos with an increase in the fetus [221]. FMO3 abundance in human liver microsomes demonstrated a 2.2-fold increase (13.0 ± 11.4 pmol/mg protein vs. 28.0 ± 11.8 pmol/mg protein) from neonates to adults with achievement of adult values after 6 years of age [222].

In neonates, the levels of protein expression for alcohol and aldehyde dehydrogenases were detected to be lower than in adults: 3-fold for ADH1A, 8-fold for ADH1B, 146-fold for ADH1C, and 3-fold for ALDH1A1. An increase in the levels was seen during the first year of life, and adult values were reached between 1 to 6 years. An exception was detected for ADH1A, its protein abundance in adults was ~40% lower than in the early childhood group [223].

Phase II reactions are catalyzed by uridine diphosphate glucuronosyltransferase (UGT), sulfotransferase (SUT), glutathione-S-transferase (GST), and N-acetyltransferase (NAT) families.

Sulfation of xenobiotics represents one of the most common phase II reactions. It aims to increase the hydrophilicity of drugs, enabling their biliary excretion and subsequent renal clearance. Sulfotransferase age-dependent activity was estimated in several studies. The activity of the cytosolic SUT measured with 2-naphthol revealed a mean value of 0.18 nmol/min/mg protein in fetal samples and 0.63 nmol/min/mg protein in adults [224]. In the study by Ladumor MK et al. (2019) significant age-dependent protein abundance was demonstrated for SULT1A1, SULT1B1, and SULT2A1, whereas SULT1A3 was stable across 0–70 years. In neonatal samples, mean abundances of SULT1A1, SULT1B1, and SULT2A1 accounted for 24%, 19%, and 38% of the adult values. The highest abundance during early childhood (1 to <6 years) was defined for SULT1A1 and SULT2A1 [225].

Glucuronidation is aimed at the completion of sulfation, and conjugation with glucuronic acid occurs at the same molecular moieties as sulfation. As a result of glucuronidation, inactive forms are produced with subsequent excretion of bile or urine. The phenomenon of neonatal jaundice suggested that UGT activity may be absent in the first days of life resulting in severe unconjugated bilirubinemia. Miyagi SJ et al. (2011) demonstrated minimum activity levels of UGT1A1 at birth with an increase of up to 90% of maximum values at 3.8 months of age. For UGT1A6 90% maximum activity was seen at 14 months of age. And for both enzymes estimation of protein expression revealed no age dependence [226].

Glutathione conjugation is specific for potent electrophiles, substrates include electrophilic carbon, nitrogen, oxygen, and sulfur atoms. Reaction products are inactive substances. GST isoenzymes demonstrated some age-dependent changes. Protein levels of GSTA1 were reported to be the lowest in the fetus (10–42 weeks) with a peak in the perinatal period (2–85 weeks) and a secondary decrease in adults. GSTA2 levels demonstrated a gradual increase from minimum in fetal samples to maximum in adults. GSTM levels were the lowest in the fetus with an increase of nearly up to adult values in the perinatal period. And opposite age distribution was revealed for GSTP—maximum in the fetal period with a decline to minimum in adults [227]. High levels of serum GST class Pi were detected in premature infants with severe perinatal asphyxia, compared to those with moderate asphyxia. Estimated high levels of serum GST Pi in the first 6 h after birth were associated with increased mortality and development of acute kidney failure in preterm neonates [228].

Considering antibacterial agents, it is important to discuss acetylation reactions, which involve compounds with amino or hydrazine residues, like sulfonamides or isoniazid. N-acetyltransferase age-dependent changes may result in variable efficacy and safety of these drugs. Insufficient or slow acetylation by NAT2 of sulfamethoxazole may result in increased toxicity risk including idiosyncratic adverse drug reactions [178]. NAT2 is also an enzyme catalyzing acetylation of the first-line antituberculosis agent isoniazid. Age was estimated as a critical factor for activity and affinity of NAT2, and the maximal adult type of activity was encountered after 5.3 years old [229].

### 2.4. Excretion

Drugs are excreted mainly with urine or bile, and total drug clearance is a sum of renal and hepatics clearances. Excretion with bile is specific for parenterally administered hydrophilic drugs extensively bound to plasma proteins, for orally administered drugs highly bound to hepatic transport proteins, and, in general, for polar large molecules [230]. Antibiotics excreted predominantly with bile include macrolides, rifampicin, fluoroquinolones, oxazolidinones, and lincosamides. Developmental changes in bile excretion were demonstrated in the work by Johnson TN et al. (2016) [231]. For neonates (26-week gestational age) a fraction of adult BE activity of 15% was necessary to predict the clearance of azithromycin, and 100% activity was apparent by 7 months. In premature neonates, (30 weeks gestational age) decreased biliary excretion was demonstrated for digoxin (10% of adults value). The ontogeny of biliary excretion reported for azithromycin, ceftriaxone, digoxin, and buprenorphine revealed rapid achievement of adult values, from days to several months after birth [231].

Antibiotics with high rates of urinary excretion include penicillins, cephalosporines, carbapenems, monobactam aztreonam, aminoglycosides, glycopeptides, cyclic lipopeptide daptomycin, phosphonic antibiotic fosfomycin. The value of renal clearance is based on the sum of the next renal functions: glomerular filtration, tubular secretion, and tubular reabsorption. Developmental changes are seen both in the structure and function of kidneys [232]. Nephrogenesis starts from the 6th week of gestation and its completion happens in the 36th week [233]. In preterm infants, nephrogenesis may continue for up to 40 days after birth. In preterms, kidneys are smaller than in term ones and a doubling of the mean total kidney volume happens from 28 to 37 weeks (from 10.3 cm^3^ at 28 weeks postmenstrual age to 19.2 cm^3^ at 37 weeks), though the estimated glomerular filtration rate (eGFR) between premature neonates and term neonates revealed no significant differences (43.5 [39.7–48.9] vs. 42.0 [38.2–50.0] mL/min/1.73 m^2^) [234]. Further study reported that infants born preterm were characterized by a smaller total kidney volume even at 24 months of age compared with term ones (56.1 (9.4) vs. 64.8 (10.2) mL), and by the same values of eGFR [235]. The meta-analysis by Smeets NJL et al. (2022) included 978 measured GFR values from 881 healthy, term-born neonates. Results revealed a doubling of GFR in the first 5 days of life (from 19.6 (95% CI, 14.7 to 24.6) to 40.6 (95% CI, 36.7 to 44.5) mL/min per 1.73 m^2^) followed by a gradual increase with the achievement of 59.4 (95% CI, 45.9 to 72.9) mL/min per 1.73 m^2^ at 4 weeks of age [236].

Salem F et al. (2021) developed an ontogeny model for GFR emphasizing the potent impact of birth on GFR value: longer period outside the uterus (greater postnatal age) resulted in higher GFR values in neonates with the same postmenstrual age. The authors stated that the difference in GFR between pre-term and full-term neonates with the same postmenstrual age is eliminated from beyond 1.25 years [237]. The evaluation of molecules related to glomerular function in preterm neonates (*n*—40, mean gestational age 30 ± 1 weeks) revealed a pronounced increase in GFR from 72 h until 3 weeks of life [238].

Generally, GFR is increased depending on gestational and post-menstrual age achieving adult values by 1-year post-natal age, and during this maturation period, the effective filtration pressure is driven by the interaction between multiple endogenous intra-renal vasoconstrictive and vasodilator factors [239]. A pathological shift of GFR in neonates may be mediated by a number of pathological states, including hemodynamic disturbances, very low-birth weight, and external factors (use of drugs resulting in vasoconstriction and nephrotoxicity), as it is shown on the Figure 2.

ACEi—angiotensin-converting enzyme inhibitorsANP—atrial natriuretic peptideAT I, AT II—angiotensin I and II NO—nitric oxidePGI2, PGE2—prostaglandins I2 and E2.

The next categories of neonates in NICU are at high risk of acute kidney injury [240]:-Premature and low birth weight neonates;-Neonates with congenital heart disease and cardiac surgery;-Neonates with hypoxic ischemic encephalopathy;-Neonates with necrotizing enterocolitis;-Neonates on extracorporeal life support;-Neonates using nephrotoxic drugs.

Antibacterials with possible negative effects on renal function may be involved in the development of tubulointerstitial nephritis (penicillins, cephalosporins, fluoroquinolones, sulfonamides) or acute tubular necrosis (aminoglycosides and amphotericin B) in neonates. The work by Rhone ET et al. (2014) revealed that for very low birth weight infants exposure to ≥1 nephrotoxic drug was reported in 87% (gentamicin—86%, indomethacin—43%, vancomycin—25%) [241].

The accurate estimation of GFR in neonates is essential for ICU treatment, increasing its effectiveness and decreasing the risk of negative outcomes. Methods of GFR determining include measurement of clearance of creatinine, inulin, iohexol, 51Cr-EDTA (ethylene diamine tetra-acetic acid), and 99mTc—DMSA (dimercaptosuccinic acid). The GFR estimate in children is mainly performed with updated versions of the Schwartz equation, though the serum cystatin C level may be used together with the serum creatinine level to estimate GFR [242]. Population pharmacokinetics modeling developed to predict clearance of renally eliminated drugs in neonates with normal renal function should be based on the body weight and postmenstrual age [243].

Developmental changes in renal function can affect the excretion of antibiotics, altering the value of clearance (CL) and, therefore, antibiotic exposure.

Table 9 demonstrates comparative data on PPB, *Vd,* and CL of antibiotics in neonates and adults.

## 3. Antibiotic Dosing Regimens in Neonates

Lack of pharmacokinetic trials and RCT including neonates results in challenges in antibiotics dosing; underdosing and underdosing are common factors of pharmacotherapy failure. Available data on antibiotics PK parameters together with simulation studies help to determine proper regimens for antibiotics used in neonates. Information about the proposed regimens of antibiotic dosing in neonates is given in Table 10.

## 4. Conclusions

Bacterial infections in neonates are considered to be among the main causes of NICU admission and may be life-threatening, especially to preterm newborns. NS highly contributes to neonatal mortality [320] emphasizing the importance of the proper antibiotic choice and dosing. Optimal dosing in neonates is essential to achieve the eradication of bacteria, prevent antibiotic resistance, and reduce the risk of possible toxic effects. There is a significant difference between preterm and full-term newborns, resulting in the inability to apply the same treatment schemes. Different rates of maturation of organs and tissues, transport proteins, and enzymes lead to the variability of PK parameters in heterogenous neonatal populations. Gastric emptying and renal, hepatic, and intestinal functions are potent factors affecting the absorption, distribution, metabolism, and excretion of antibiotics. Current data actualize and transform the postulate “children are not small adults” into “preterm neonates are not small full-term ones”. The extrapolation of PK data from more mature human subjects to neonates, and from term neonates on preterm neonates is challenging, making actual the need for additional pharmacokinetic studies, pharmacogenetic studies (distinguishing genetic and non-genetic factors affecting PK parameters of antibiotics in neonates [321,322]) and RCT in this vulnerable category of patients.

## Figures and Tables

**Figure 1 biomedicines-11-00940-f001:**
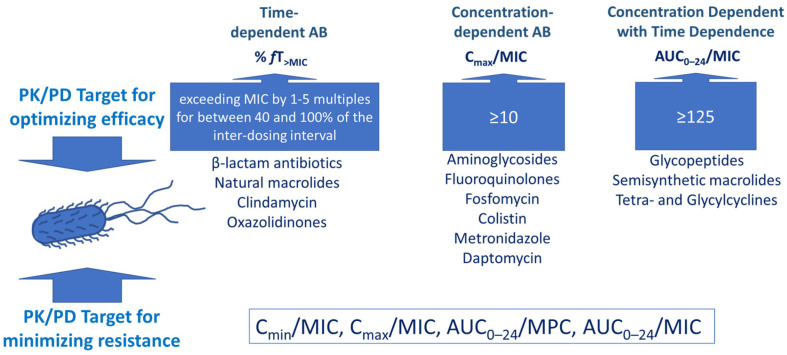
PK/PD targets for optimizing efficacy and minimizing antibacterial resistance for time-dependent, concentration-dependent, and concentration-dependent with time dependence antibiotics.

**Figure 2 biomedicines-11-00940-f002:**
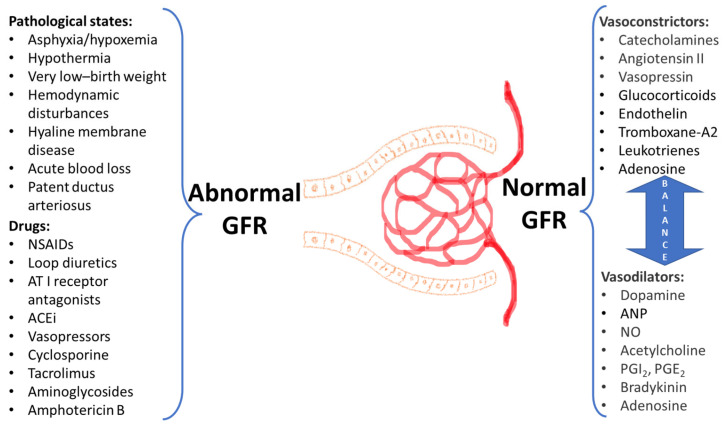
Factors affecting GFR in neonates.

**Table 1 biomedicines-11-00940-t001:** Time of expression, age-associated changes, and antibiotic-substrates for the main intestinal transport proteins.

Transport Protein	Time of Expression in Different Cells	Age-Dependent Change of Expression	Transported Antibiotics
BCRP	Intestinal epithelium—from 5.5 to 28 weeks of gestation. Hepatocytes—from 10 to 11 weeks [45]	Similar levels in preterm newborns, full-terms, and adults [43,46]	Delafloxacin, ciprofloxacin, enrofloxacin, nitrofurantoin, norfloxacin, ofloxacin [47,48]
P-gp	Enterocytes—from the 12 weeks of gestation [45]	Expression in fetus is lower than in adult samples [43]	Erythromycin, tetracycline [49] Azithromycin [50] Levofloxacin, sparfloxacin [51] Dicloxacillin [52,53]
MRP2	Appears in the liver of 14-week-old fetuses, strong expression at 19 weeks [54]	Lower protein expression in fetuses and term newborns than in adults [43] MRP2 mRNA was 30-fold lower in fetal, 200-fold lower in neonatal, and 100-fold lower in infant liver compared to adults [55]	Ampicillin, azithromycin, and ceftriaxone, cefodizime, ceftriaxone [44,56]
OATP1A2	-	-	Levofloxacin [49] Ciprofloxacin, enoxacin, gatifloxacin, levofloxacin, lomefloxacin, norfloxacin [57] Erythromycin [58] Tebipenem [59]
OATP1B1	-	High expression in the fetus and low expression in the term newborns, with stable protein levels further on [43] OATP1B1 expression was 20-fold lower in fetal, 500-fold lower in neonatal, and 90-fold lower in infant liver compared to adults [55]	Benzylpenicillin, rifampicin, rifampin, rifampicin [49] Cefazolin, cefditoren, cefoperazone, nafcillin [60]
OATP1B3	-	OATP1B3 mRNA was 30-fold lower in fetal, 600-fold lower in neonatal, and 100-fold lower in infant liver compared to adults [55] OATP1B3 exhibited high expression at birth, decline over the first months of life, and then increase in the preadolescent period [61].	Rifampicin, rifampin [49] Cefadroxil, cefazolin, cefditoren, cefmetazole, cefoperazone, cephalexin, nafcillin [60] Erythromycin [58]
OATP2B1	-	Similar levels in samples from all age groups [43] Intestinal OATP2B1 expression in neonates was significantly higher than in adults [55]	Benzylpenicillin [62] Tebipenem pivoxil [63]
PEPT1	Enterocytes—from 24.7 to 40th week of gestation (comparable level with full-terms) [43]	The PEPT1 mRNA expression of the neonates/infants was only lightly lower (0.8-fold) than the older children (*p* < 0.05) [55]	Cefadroxil, ceftibuten, cefixime, cephradine, cephalexin [60]

**Table 2 biomedicines-11-00940-t002:** The summary of GIT features in neonates and their effects on absorption and bioavailability of orally administered drugs.

Parameter	Age-Associated Change	Effect on Absorption and Bioavailability
pH	Value close to neutral at birth was described in some studies with rapid change to acidic values	High gastric pH may provide positive effect on the oral bioavailability of acid-labile drugs (e.g., ampicillin, amoxicillin, penicillin, nafcillin, erythromycin), it favors the ionization, and reduces the absorption of weak acids and may negatively affect absorption of weak bases by decreasing their solubility [64]
Gastric volume	Decreased	Solubility of some drugs can be decreased
Gastric emptying and intestinal transit	Some slowing is specific, intestinal transit in preterm newborns is 4 times longer than in adults [37]	Delay and decrease in absorption
Biliary function	Decreased compared with older children and adults [31]	Possible decrease in the intestinal absorption of lipid-soluble drugs
Pancreatic enzymes	Decrease at birth with further rise through the first year of life	Decreased intra-duodenal hydrolysis may result in incomplete absorption
Intestinal surface area	Reduced surface-to-volume ratio in children when compared with adults [64]	Reduced absorptive surface may influence absorption of some drugs
Intestinal permeability	In preterm infants (26–36 weeks gestation), intestinal permeability is higher than in healthy term infants only if measured within two days of birth [65]	Increase in absorption may by for drugs with paracellular route of absorption [66]
Transport proteins	Decreased p-gp, MRP1, MRP2—decreased efflux function	Possible increase in absorption of corresponding substrates
	Light decrease in PEPT1	Possible decrease in absorption of corresponding substrates
Intestinal wall drug-metabolizing enzymes	Possible decrease in cytochrome P450 isoenzime 3A4 (CYP3A4) in neonates at birth with further rise through childhood period [67]	Possible increase in absorption and bioavailability of CYP3A4 substrates

**Table 3 biomedicines-11-00940-t003:** Food effects on absorption of oral antibiotics in pediatric and adult populations.

Drug (Oral Administration)	Food Effect on PK Parameters in Pediatric Population	Food Effect on PK Parameters in Adults
Penicillin V (phenoxymethylpenicillin)	*C_max_* decreased	Penicillin V (phenoxymethylpenicillin)
Amoxicillin	*C_max_* decreased at lower doses, unchanged at higher doses *AUC* unchanged at all doses	No effect
Ampicillin	*C_max_* unchanged *AUC* unchanged	Some studies suggest reduction in *C_max_* and *AUC* with food
Cefpodoxime proxetil	*T_max_* prolonged *C_max_* unchanged	*C_max_* and *AUC* increase with any meal intake
Cephalexin	*C_max_* lightly decreased *AUC* lightly increased	Delay in absorption *AUC* unchanged
Cefaclor	*C_max_* decreased	*T_max_* prolonged *C_max_* decreased
Cefadroxil	*C_max_* decreased	No effect
Cephradine	*C_max_* decreased	No effect
Cefixime	No clinically significant changes in *C_max_* and *AUC*	*T_max_* prolonged *C_max_* and *AUC* unchanged
Clarithromycin	*C_max_* unchanged *AUC* unchanged	*C_max_* unchanged *AUC* unchanged

**Table 4 biomedicines-11-00940-t004:** Age-associated change of absorption for non-oral routes of drugs administration.

Routes of Administration	Main Physiological Factors Affecting PK Parameters in Neonates	Change of Absorption
Intramuscular	Blood flow to muscle: variable decrease over the first 2–3 weeks of life. The ratio of muscle mass to body mass: less in neonates than in adults Water content: higher proportion of water in neonates [118]	Possible variability in absorption, though for many antibiotics it nearly reaches adult values. Hydrophilic drugs may have greater intramuscular absorption in neonates than children or adults due to high water content in muscles
Rectal	pH: decreased in neonates compared with older infants (mean pH 6.47 vs. 6.90) [119] and with adults (pH 7.2–7.4) [120]	Nearly no change in absorption of drugs (absorbable drugs should have *pKa* values near or above the physiological range)
Percutaneous	Skin thickness and keratinization: reduced Hydration of the stratum corneum: increased Surface area to bodyweight ratio: higher than in adults [38,121]	Increased absorption of drugs including some topical antibiotics with risk of overexposure, and potential toxic effects
Inhalation Intranasal	Stage of lung development: late preterm neonates (28–34 weeks of gestation) may have saccular stage of lung development, more preterm infants—just the stage of development of bronchioles and alveolar epithelium, both with surfactant deficiency and high risks for respiratory distress syndrome [122] Permeability of the mucosa of the nasal cavity: Increased in neonates [121]	Reduced absorption from the lower respiratory department and increased absorption from the upper respiratory department including nasal cavity (e.g., with facemask for inhalation)

**Table 5 biomedicines-11-00940-t005:** Relative rates of antibiotics distribution to the different organs and tissues.

Antibiotic	CNS	Lungs	Gallbladder, Bile	Urinary Tract	Bones and Joints
Fosfomycin	+++	++	++	+++	++
Lincosamides	+ High doses may result in therapeutic concentrations achievement	++	++	+/++	++/+++
Rifamycins	+++	+++	+ (?)	+/++	++/+++
Fluoroquinolones	++/+++ Minimum values for ciprofloxacin, higher value for respiratory fluoroquinolones	+++ High in alveolar cells, inflammation may decrease penetration	+/++ Higher for levofloxacin, ofloxacin, ciprofloxacin	++/+++	+++
Macrolides	+ Rapid removement by P-gp of BBB	+++ Higher values in the lung parenchyma, less in epithelial lining fluid	−/+	−/+	++
Aminoglycosides	+ Penetration may be increased in preterm infants with increased risks of aminoglycoside-dependent ototoxicity	++ Only in epithelial lining fluid and bronchial mucus, no cellular entry	−/+	+++	++
Glycopeptides	? Variable data, not predicting clinical outcome, penetration may increase in neonates with meningitis	+++	++/+++	+++	++/+++
Lipoglycopeptides	+ Poor penetration	+++	+ (?)	++/+++	+++
Cephalosporines	++/+++ Especially high for ceftriaxone, cefotaxime, ceftazidime, cefixime and cefepime Low—for cefuroxime. Meningeal inflammation increases penetration	+++ High concentrations in the lung parenchyma, epithelial lining fluid and bronchial mucus	++/+++ The highest value is for cefoperazone, cefotetan, minimum—for cefotaxime	+++	++/+++
Penicillins	+ Negligible with non-inflamed BBB and increased in meningitis. Dose increase is needed in CNS infections	++/+++ High in the epithelial lining fluid and bronchial mucus	++/+++ The highest value is for piperacillin/tazobactam	+++ High concentrations in urine	−/+ Undetectable, exception for flucloxacillin, piperacillin tazobactam, amoxicillin clavulanat
Carbapenems	+++ Imipenem reaches higher CSF concentrations, but causes seizures in neonates, especially preterms, thus meropenem is preferred	++/+++	+ Despite low bile concentrations, some studies demonstrated clinical efficacy of imipenem and meropenem)	+++ Imipenem: adding cilastatin prevents degradation with dehydropeptidase-I in kidneys	++/+++
Oxazolidinones	++ Penetration in CNS tissues and CSF, brain concentration of linezolid is higher than for tedizolid	+++ Concentrations in epithelial lining fluid > serum > alveolar macrophages)	++ (?)	++	+++

?—more studies are needed; + detectible concentrations (low levels); ++ moderate concentrations; +++ high concentrations.

**Table 6 biomedicines-11-00940-t006:** Antibacterials with different bilirubin displacing activity (Reprinted with permission from [175] Copyright year, 2023; Copyright owner, Elsevier).

Moderate to Strong Displacing Effect	Minimal Displacing Effect	No Displacing Effect
Sulfamethoxazole Sulfisoxazole Sulfadiazine Ceftriaxone Cefotetan Moxalactam Dicloxacillin Carbenicillin Cefazolin	Nafcillin Ampicillin Oxacillin Methicillin Cefotaxime	Aminoglycosides

**Table 7 biomedicines-11-00940-t007:** CYP450 isoenzymes ontogeny and antibiotics-substrates.

CYP450 Isoenzyme	Antibiotic-Substrate	Comment on Ontogeny
CYP1A1	Linezolid [186]	Absent-to-low expression in the neonate; activity reaches 50% of adult values by 1 year of age [193]
CYP1A2	-	Becomes active at 1 to 3 months [194] The median expression was significantly greater after 15 months postnatal age (*n* = 55) than in fetal and younger postnatal samples [195]
CYP2A6	-	Low activity during fetal age (<1% of adult values) with increase up to 50% adult activity values at neonatal age. Activity and protein expression reach adult values between 1 and 15 years of age [196]
CYP1B1	Linezolid [186]	Appears in the early stages in ontogenesis [197]
CYP2B6	-	2-fold increase in expression and activity in the first month of life, it is suggested that there are no additional changes between 1 month and 18 years or a 7-fold increase between 1 year and adulthood [198]
CYP2C8	Linezolid [186]	The median expression was significantly greater among samples from subjects older than 35 postnatal days (*n* = 122) compared with fetal samples and very young infants (fetal to 35 days postnatal, *n* = 100) [195]
CYP2C9	-	Expression at 1 to 2% of mature values during the first trimester, with progressive increases during the second and third trimesters to levels approximately 30% of adults. From birth to 5 months values varied 35-fold and were significantly higher than those observed during the late fetal period, with 51% of samples exhibiting values commensurate with mature levels. Less variability was observed between 5 months and 18 years [199]
CYP2C19	-	Expression did not change at birth, increased linearly over the first 5 postnatal months, and varied 21-fold from 5 months to 10 years [199]
CYP2D6	Linezolid [186]	Hepatic expression in neonates <7 days was higher than in first and second trimesters, but close to the values of third trimester. Expression in neonates >7 days was higher than in earlier ages [200] Adult activity is achieved by 3–5 years of age [193]
CYP2E1	-	Lower value in neonatal samples compared with infants 31 to 90 days of age, increased with age. Median (range) values of microsomal protein for older infants, children, and young adults were 8.8 (0–70); 23.8 (10–43); 41.4 (18–95) pmol/mg, respectively [201]
CYP3A4	Erythromycin [156] Linezolid [186] Clindamycin [202]	Activity is extremely low in the fetus; it begins to increase after birth reaching 30–40% of the adult value after one month, 50% at 6–12 months, and 100% at age 1–5 years [203] Protein expression was 200-fold lower in the fetal tissues compared to either the pediatric or adult tissues [191]
CYP3A5	Linezolid [186] Clindamycin [111]	Protein expression is relatively stable (0.001–0.005 nmol/mg) in fetal, pediatric, and adult samples [191]
CYP3A7	Clarithromycin [204]	Highly active in fetuses with a subsequent decrease in activity after first week of life, reaching minimal levels in adults [204] Protein expression was 10-fold higher in the fetal tissues compared to either the pediatric or adult tissues [191]

**Table 8 biomedicines-11-00940-t008:** Effect of antibiotics on human CYP450 enzymes (direct and indirect).

CYP450 Isoenzyme	Antibiotics-Inhibitors	Antibiotics-Inducers
CYP1A2	Ciprofloxacin [207] Caderofloxacin, antofloxacin, moxifloxacin, gatifloxacin * [208]. Oritavancin * [214]	-
CYP2B6	Oritavancin * [214]	Rifampicin [215]
CYP2C8	Piperacillin [211]	Rifampicin [215] Trimethoprim [216,217]
CYP2C9	Oritavancin * [214].	Rifampicin [215] Sulfamethoxazole [216] Dicloxacillin [53] Levofloxacin [208]
CYP2C19	Oritavancin * [214] Meropenem [213]	Rifampicin [215] Dicloxacillin [53]
CYP2D6	Oritavancin * [214]	Oritavancin [214]
CYP3A4	Erythromycin, roxithromycin, clarithromycin, troleandomycin, telithromycin, josamycin [205] Oritavancin * [214] Meropenem [213]	Flucloxacillin [212] Dicloxacillin [53] Rifampin (inducer of CYP 3A4) [206,215] Oritavancin * [214]
CYP3A5	Erythromycin, clarithromycin [205]	-
CYP3A7	Erythromycin [218]	Rifampicin, troleandomycin, erythromycin (upregulation of expression) [218,219]

* Negligible effect.

**Table 9 biomedicines-11-00940-t009:** PPB, *Vd* and CL of antibiotics in neonates and adults.

Drug	*Vd*		PPB Rate		*CL*		*T1* */2*	
	Adults	Neonates	Adults	Neonates	Adults	Neonates	Adults	Neonates
Penicillin Group								
Penicillin G	0.53–0.67 L/kg [244]	Extremely preterm neonates comparable with full-terms, mean value 0.50 L/kg, [245]	45–68% [244]	60% [245]	560mL/min [244]	Median—0.21 L/h (interquartile range, 0.16 to 0.29 L/kg) [245]	0.4–0.9 h [244]	Median—3.6 h (interquartile range, 3.2 to 4.3 h) [245]
Ampicillin and Ampicillin sulbactam	14.2 to 15.1 L [246]	GA 22–24 weeks: 0.25 L/kg; GA 25–27 weeks: 0.34 L/kg [247]	20% [248]	Some decrease is proposed [247]	10.7 L/h [249]	GA 22–24 weeks: 0.023 L/h/kg; GA 22–24 weeks: 0.036 L/h/kg [247]	1 h [246]	GA 22–24 weeks: 7.6 [6.4–9.0] h; GA 25–27 weeks: 6.6 [5.6–7.9] h [247]
Amoxicillin	27.7 L [250]	GA 25 to 42 weeks: 0.65 L/kg [251];	<20% [252]	Less than in adults [252]	21.3 L/h [250]	0.096 ± 0.036 L/kg^−1^ h^−1^ [251]	~1 h [250]	5.2 ± 1.9 h [251]
Flucloxacillin	About 10 L [253]	GA 26 to 42 weeks: 0.54 ± 0.17 L/kg [254]	95–96% [124]	GA 25 to 41 weeks: 86.3% [139]	93.1 mL/min [255]	GA 26 to 42 weeks: 0.18 L/kg/h [254]	0.88 h [255]	2.6 ± 1.6 hours [254]
Piperacillin	0.27 ± 0.13 L/kg [256]	PNA < 14 weeks: 0.42 L/kg [257]	20–30% [256]	Median GA of 30 weeks: 30% [257]	0.183 L/h/kg [258]	Median GA of 30 weeks: median value of 0.085 L/h/kg [257]	1.1 ± 0.6 h [256]	3.5 (1.7–8.9) h [257]
Ticarcillin	15.7 L [258]	Preterms > 30 weeks of gestation: 0.26–0.34 L/kg Fullterms: 0.22–0.23 L/kg Total population 0.48 L/kg [259]	45–65% [258,260]	Less than 45% [259]	46.2 ± 10.9 mL/min/m^2^ [260]	PNA < 14 days: 0.078 L/kg/h; PNA ≥ 4–45 days: 0.078 L/kg/h [259]	1.23 h [258]	PNA < 14 days: 24.3 (11.2–35.7) h; PNA = 14 days: 15.1 (5.0–44.8) h [259]
Cephalosporins								
Cefotaxime	33.3 L/1.73 m^2^ [261]	GA 30 to 40.1 weeks: median value of 0.36 L/kg [262]	Median value of 29.45–29.86% in critically ill adults against 27–38% in healthy volunteers [261,263]	GA 30 to 40.1 weeks: median value of 40% [262]	138.4 mL/min [261]	0.04 (0.02–0.09) L/h/kg [262]	1.1 hours [261]	GA < 32 weeks, PNA < 7 days: 3.49 ± 0.45 h, PNA ≥ 7 days: 3.72 ± 0.85 h; GA 32–63 weeks, PNA < 7 days: 3.28 ± 0.35 h, PNA ≥ 7 days: 2.0 ± 0.24 h; GA ≥ 37 weeks, PNA < 7 days: 2.77 ± 0.49 h, PNA ≥ 7 days: 2.01 ± 0.57 h [264]
Cefepime	0.2 L/kg [265]	Mean GA 38.1 weeks: median value of 0.62 L/kg [266]	20% [265]	Mean GA 38.1 weeks: ≤20% [266]	7.2 ± 0.48 L/h [265]	0.18 (0.13–0.24) L/h/kg [266]	2–2.3 h [265]	4.9 ± 2.1 h [267]
Carbapenems								
Ertapenem	0.7 L [268]	0.4–0.6 L/kg [269]	92–95% [146]	Some decrease is proposed	0.7 L/h [268]	0.05–0.25 L/h/kg [269]	6.1 h [268]	NA
Meropenem	21 L [270]	GA 23–41 weeks: 0.4 L/kg [271]	2% [271]	NA	27.6–36.3 mL/min [146]	GA 23–41 weeks: 0.104 L/h/kg [271] 0.093–0.238 L/h/kg [269]	1 h [270]	PNA 5–44 days: 1.3–5.4 h [269]
Imipenem	0.22 L/kg [272]	GA 24–41 weeks: 0.73 L [272]	20% [272]	<20% [272]	0.2 L/h/kg [273]	0.21 L/h [272]	1–3 h [272]	1.9 h in term neonates, 2.4 h in preterm neonates [272]
Doripenem	16.8 L (8.09–55.5) [274]	GA <32 weeks: ~0.56 L/kg Infants: ~0.55 L/kg [275]	8% [274]	NA	16.0 L/h [274]	PNA < 4 weeks: 2.03 mL/min/kg; PNA > 4 weeks: 3.03 mL/min/kg [275]	1 h [274]	2.98 h [275]
Macrolides								
Azithromycin	31.1 L/kg [276]	GA 24–28 weeks: 1.88 L/kg [277]	30% [158]	NA [158]	37.8 L/h [276]	0.15 L/h/kg^0.75^ [277]	68 h [276]	58 h [159]
Oxazolidinones								
Linezolid	In most studies 50–60 L [278]	GA < 34 weeks: 0.92 L/kg; GA ≥ 34 weeks: 0.72 L/kg [279] Preterms < 1 week: 0.81 L/kg; Fullterms < 1 week: 0.78 L/kg; Fullterms ≥ 1 week ≤ 28 days: 0.66 L/kg [280]	10.5–31% [278,280]	NA	5.54 L/h for 30-year-old patients; 1.35 L/h for 85-year-old patients [281]	GA < 34 weeks: 0.17 (0.07–0.31) L/h/kg; GA ≥ 34 weeks: 0.16 (0.14–0.19) L/h/kg [279] Preterms < 1 week PNA: 2.0 mL/min/kg; Fullterms < 1 week PNA: 3.8 mL/min/kg; Fullterms ≥ 1 week ≤28 days: 5.1 mL/min/kg [280]	4–6 h [280]	Preterms <1 week PNA: 5.6 [2.4–9.8] h; Fullterms <1 week: 3.0 [1.3–6.1] h; Fullterm ≥1 week ≤28 days: 1.5 [1.2–1.9] [280]
Lipoglycopeptides								
Telavancin	0.1–0.15 L/kg [282]	NA	90–93% [282]	NA	0.012–0.014 L/h/kg [282]	NA	6–8 h [282]	NA
Lipopeptides								
Daptomycin	0.1 L/kg [283]	GA 23–40 weeks: 262.4 mg × h/L [284]	>90% [283]	NA	0.01 L/h/kg [283]	GA 23–40 weeks: 0.21 L/kg [284]	8–9 h [282]	NA
Fluoroquinolones								
Levofloxacin	1.09–1.26 L/kg [169]	0.5–2 years old: 1.56 L/kg (IV), 2.32 L/kg (PO) [103]	24–38% [115]	NA	0.5–2 years old: 0.35 L/h/kg (IV), 0.31 L/h/kg (PO) [103]	5.76–8.52 L/h [169]	6–8 h [169]	0.5 to <2 years: 4.1 ± 1.3 h [103]
Moxifloxacin	2.1 L/kg [285]	2.25 L/kg [285]	39% [285]	NA	12 ± 2 L/h [171]	0.35 L/h/kg [285]	11.5–15.6 h [171]	8.6 h [286]
Aminoglycosides								
Amikacin	24 L [287]	0.47–0.61 L/kg [288]	≤10% [287]	NA	100 mL/min—serum clearance 94mL/min—renal clearance [287]	0.025–0.097 L/kg/hr [288]	2–3 h [287]	PMA ≤ 29 weeks, PNA 0–7 days: 17.6 ± 4.8 h, PNA ≥ 8 days: 9.8 ± 2.1 h; PMA 30–33 weeks, PNA 0–7 days: 12.3 ± 0.5 h, PNA ≥ 8 days: 7.3 ± 3.2 h; PMA 34–36 weeks, PNA 0–7 days: 6.3 ± 1.4 h, PNA ≥ 8 days: 5.2 ± 1.6 h. [288]
Lincosamides								
Clindamycin	0.79 L/kg [289]	PMA < 28 weeks: 1.20 L/kg; PMA > 28–32 weeks: 1.3 L/kg; PMA > 32–40 weeks: 1.03 L/kg; PMA > 40–60 weeks: 0.99 L/kg [165]	78–94% [289]	81% [164]	0.3–0.4 L/h/kg [289]	PMA < 28 weeks: 0.14 L/kg/h; PMA > 28–32 weeks: 0.19 L/kg/h; PMA > 32–40 weeks: 0.24 L/kg/h; PMA > 40–60 weeks: 0.28 L/kg/h [164]	2.4 h [202]	PMA ≤ 38 weeks: 5.89 (2.42–12.90) h; PMA > 28–32 weeks: 5.25 (2.34–8.87) h; PMA > 32–40 weeks: 3.96 (1.30–8.83) h; PMA > 40–60 weeks: 2.35 (0.94–6.44) h; PMA > 5 months < 1 year: 2.05 (1.26–3.47) h [164]
Glycopeptides								
Teicoplanin	0.86 L/kg [290]	PMA 26 to 43 weeks: 0.283 L [291];	90–95% [149]	80.5–71.9% [150]	0.0114 L/h/kg [290]	0.0227 L/h [291]	70–100 h [149]	29.75 ± 8.86 h [291]
Vancomycin	0.4–1 L/kg [292]	Preterms, median age 14 (3–58) days: 0.884 L [293]; GA 23–34 weeks: 0.54 ± 0.10 L/kg [294]	10–50% [292]	NA	2.64 L/h [292]	Preterms, median age 14 (3–58) days: 0.1 L/h [293]; GA 23–34 weeks: 0.06 L/h/kg [294]	4–12 h [283]	3.5–10 h [295]
Other groups of antibiotics								
Rifampin	1.6 L/kg [117]	1.84 ± 0.59 [296]	72–92% [173]	NA	10–19.3 L/h [117]	3.6 L/h, 70 kg [117]	2.5 h [117]	PNA < 14 days: 7.1 (3–23.9) h; PNA ≥ 14 days: 3.5 (1.9–6.5) h [117]
Fosfomycin	0.42 L/kg [113]	GA 38–40 weeks: 0.38 L/kg [172]	Negligible [172]	NA	8.7 ± 1.7 L/h [113]	0.14 L/h [172]	2.4–2.8 h [113]	2.4 ± 0.4 h [113]
Metronidazole	0.25–0.85 L/kg [297]	0.71 L/kg [297]	20% [297]	NA	GA < 26 weeks: 0.024 (0.010–0.086) L/h; GA 26–29 weeks: 0.026 (0.012–0.076) L/h; GA 30–32 weeks: 0.029 (0.015–0.074) L/h [297]	6–10 h [298]	8 h [297]	GA < 26 weeks: 20.5 (5.7–49.9) h; GA 26–29 weeks: 18.6 (6.5–38.7) h; GA 30–32 weeks: 16.7 (6.5–38.7) h [297]

GA—gestational age; PMA—postmenstrual age; PNA—postnatal age.

**Table 10 biomedicines-11-00940-t010:** Antibiotic dosing regimens in neonates.

Antibiotic	Regimen and Indications for Critically Ill Neonates	Comments	PK/PD Target	Safety
Penicillin Group				
Penicillin G [245]	EOS: 25,000 IU/kg every 12 h for all MIC values tested (up to 2 mg/L).	GA of ≥32 weeks PNA of <72 h	40% *fT > MIC* *MIC* < 2 mg/L 100% *fT > MIC* *MIC* ≤ 0.5 mg/L	100 μg/mL At high doses—risk of neurotoxicity and encephalopathy [245]
Ampicillin [299]	50 mg/kg every 8 h	PNA 1—10 days	75% *T > MIC* *MIC* ≤ 8 mg/L	140 μg/mL At high doses—risk of neurotoxicity [247]
Ampicillin [300]	50 mg/kg every 8 or 12 h	GA ≥ 32 weeks	40% *fT > MIC;* 100% *fT > MIC*	
Ampicillin [247]	EOS: A short-course ampicillin regimen: 2 doses, 50 mg/kg every 12 h	GA 22–27 weeks PNA < 7 days, BW < 1500 g	100% *fT > MIC* *MIC* ≤ 8 mg/L	
Azlocillin [301]	100 mg/kg every 8 h	GA 30.1–41.1 weeks PNA 1.0–2.0 days	70% *fT > MIC* *MIC* = 2–16 mg/L	Risk of hepatotoxicity
Amoxicillin [302]	50 mg/kg every 24 h 75 mg/kg every 24 h	GA 36–37 weeks GA 38–42 weeks	100% *T > MIC* *MIC* = 0.25–1 mg/L	Doses > 140–150 mg/L: risk of nephrotoxicity
Flucloxacillin [254]	25 mg/kg every 4 h 10 mg/kg every 6 h	For all neonates, irrespective of their GA and PNA.	40% *T > MIC* *MIC* = 0.25–2 mg/L	200 mg/L
Piperacillin/Tazobactam [257]	100 mg/kg every 8 h	PMA ≤ 30 weeks	50–75% *fT > MIC* *MIC* = 0.5–32 mg/L	157.2 mg/L Risk of neurotoxicity [303]
	80 mg/kg every 6 h	PMA 30–35 weeks		
	80 mg/kg every 4 h	PMA 35–49 weeks		
Ticarcillin/Clavulanic acid [259]	75 mg/kg every 12 h	PNA < 14 weeks	75% *fT > MIC* *MIC* = 0.5–16 mg/L	
	75 mg/kg every 8 h	PNA ≥ 14–45 weeks		
Cephalosporins				
Ceftoraline Fosamil [304]	6 mg/kg every 8 h	<2 months	35–44% *fT > MIC* *MIC* = 0.5–2 mg/L	High risk of diarrhea
Ceftazidime [305]	25 mg/kg every 8 h for MIC 4 mg/L	0.1–2 years	70% *fT > MIC* *MIC* = 4–8 mg/L	
	50 mg/kg every 8 h for MIC 8 mg/L			
Cefotaxime [262]	50 mg/kg every 12 h	GA 30.0–41.1 weeks PNA 1.0–3.0 days	70% *fT > MIC* *MIC* = 2 mg/L	Feeding intolerance Diarrhea [262]
Cefotaxime [306]	50 mg/kg every 12 h	PNA < 7 days	75% *fT > MIC* *MIC* = 2–4 mg/L	
	50 mg/kg every 8 h	GA < 32 weeks PNA ≥ 7 days		
	50 mg/kg every 6 h	GA ≥ 32 weeks PNA ≥ 7 days		
Cefepime [266]	50 mg/kg every 12 h for MIC 4 mg/L	PMA < 38 weeks	70% *fT > MIC* *MIC* = 4–8 mg/L	
	40 mg/kg every 8 h for MIC 8 mg/L	PMA ≥ 38 weeks		
Cefazoline [307]	25 mg/kg every 12 h	PNA ≤ 7 days BW ≤ 2000 g	60% *fT > MIC* *MIC* > 8 mg/L	
	50 mg/kg every 12 h	PNA ≤ 7 days BW > 2000 g		
	25 mg/kg every 8 h	PNA 8–28 days BW ≤ 2000 g		
	50 mg/kg every 8 h	PNA 8–28 days BW > 2000 g		
Carbapenems				
Meropenem [308]	20 mg/kg every 12 h	GA < 32 weeks PNA < 14 weeks	50% *T > MIC MIC* = 4 mg/L 75% *T > MIC MIC* = 2 mg/L	64.2 mg/L—risk of neurotoxicity 44.45 mg/L for *C_min_*—risk of nephrotoxicity [309]
	20 mg/kg every 8 h	GA < 32 weeks PNA ≥ 14 weeks GA ≥ 32 weeks PNA < 14 weeks		
	30 mg/kg every 8 h	GA ≥ 32 weeks PNA ≥ 14 weeks		
Doripenem [275]	5 mg/kg every 24 h	<8 weeks PNA < 12 weeks	70–100% *T > MIC* *MIC* = 1 mg/L	Well tolerated
	8 mg/kg every 24 h	>8 weeks PNA < 12 weeks		
Imipenem [272]	20–25 mg/kg every 6–12 h	GA 24–41 weeks PNA 2–153 days	100% *T > MIC* *MIC* = 2 mg/L	
Lipopeptides				
Daptomycin [310]	6 mg/kg every 12 h	Median GA 27 weeks; PNA 5 days	NO PK/PD STUDY	Risk of elevation of liver enzymes
Fluoroquinolones				
Levofloxacin [311]	10 mg/kg every 12 h	PMA 27–42 weeks	NO PK/PD STUDY	Risk of diarrhea, hepatotoxicity, Dysglycemia
Moxifloxacin [285]	6 mg/kg every 12 h	>3 months to <2 years	*AUC_(0–24)_* at steady state 20–60 mg·h/L *C_max_* at steady state 2–6 mg/L	Risk of individual intolerance risk of QT prolongation
Moxifloxacin [286]	5 mg/kg every 24 h	GA 25 weeks	*C_max_/MIC*: >10 *AUC_0–24_/MIC*: ≥100	
Macrolides				
Azithromycin [277]	20 mg/kg every 24 h × 3 days	GA 24–28 weeks PMA < 72 h	*AUC_24_*/*MIC_90_* of >4 h	Risk of necrotizing enterocolitis, intraventricular hemorrhage
Lincosamides				
Clindamycin [164]	5 mg/kg every 8 h	PMA ≤ 32 weeks	*AUC_ss,0–8_* = 33.8 μg·h/mL *C_ss,max_* = 7.9 μg/mL *fC_ss_,_50_* = 0.12 μg/mL	No adverse events related to clindamycin use in this study
	7 mg/kg every 8 h	PMA 32–40 weeks		
	9 mg/kg every 8 h	PMA > 40–60 weeks		
Aminoglycosides				
Amikacin [312]	15 mg/kg 30–36 h	PNA ≤ 14 days BW 2000 g; ≥2800 g	*C_troug_*_h_ < 3 μg/mL *C_max_* 24 g/mL	Toxicity risk at *C_min_* > 5 μg/mL [312]
Amikacin [288]	13 mg/kg every 48 h	PNA 0–7 days PCA ≤ 29 weeks	Serum peak concentration 20–30 mg/L	
	13 mg/kg every 36 h	PNA ≥ 8 days PCA ≤ 29 weeks PNA 0–7 days PCA 30–33 weeks		
	13 mg/kg every 24 h	PNA ≥ 8 days PCA 30–33 weeks PNA 0–7 days PCA 34–36 weeks		
	13 mg/kg every 18 h	PNA ≥ 8 days PCA 34–36 weeks		
	13 mg/kg every 24 h	PNA 0–7 days PCA ≥ 37 weeks		
	13 mg/kg every 24 h	PNA ≥ 8 days PCA ≥37 weeks		
Gentamicin [313]	5 mg/kg every 48 h	Premature BW <1200 g	*C_max_*/*MIC* ratio at least 8–10	Toxicity risk at *C_min_* > 2 μg/mL and *C_max_* > 35 μg/mL Risk of nephrotoxicity, neurotoxicity, ototoxicity
	5 mg/kg every 36 h	Premature BW 1200–2500 g		
	5 mg/kg every 24 h	BW > 2500 g; Term neonates 1–28 days		
Gentamicin [314]	5 mg/kg every 36 h	PMA 30–34 weeks PNA 8–28 days PMA ≥ 35 weeks PNA 0–7 days	*C_trough_* < 1 μg/mL	
Gentamicin/Tobramicin [315]	4.5 mg/kg of gentamicin and 5.5 mg/kg of tobramycin every 72 h	PNA ≤ 5 days	*C_max_* 5–12 μg/mL *C_trough_* < 0.5 μg/mL	
Glycopeptides				
Vancomycin [293]	12 mg/kg every 8 h	GA 23–34 weeks	*AUC_0–24 h_* of ≥400 mg·h/L	>20 μg/mL—risk of nephrotoxicity [293]
Vancomycin [316]	25 mg/kg (LD) 15 mg/kg (MD) every 12 h	PMA < 35 weeks	*AUC_0–24_* of 400 mg·h/L at steady-state	
	25 mg/kg (LD) 15 mg/kg (MD) every 8 h	PMA ≥ 35 weeks		
Vancomycin [317]	32–60 mg/kg/day (4 doses)	<1 year	*AUC_0–24_*/*MIC* > 400	
	60 mg/kg/day (4 doses)	>1 year		
Vancomycin [294]	10 mg/kg every 12 h	PMA < 30 weeks	*AUC*_0–24_/*MIC* ≥ 267 at steady state	
	10 mg/kg every 8 h	PMA ≥ 35 weeks		
Teicoplanin [291]	16 mg/kg 10–11 mg/kg every 24 h	BW: <1000 g BW: 1000 <2000 g	*AUC*/*MIC* ratio of ≥400	
Other groups of antibiotics				
Rifampin [117]	8 mg/kg every 24 h	PNA < 14 days	*AUC_∞_* = 55.2 μg·h/mL *C_max,ss_* = 8.3 ± 1.8 μg/mL	Well tolerated
	15 mg/kg every 24 h	PNA 14–61 days	*AUC_∞_* = 55.2 μg·h/mL *C_max,ss_* = 13.2 ± 3.3 μg/mL	
Rifampin [296]	10 mg/kg every 24 h	GA 29.9 (4.1) weeks PNA 24.8 (13.4) days	Plasma concentrations >*MIC*	
Fosfomycin [318]	50–100 mg/kg every 8 h 50–70 mg/kg every 6 h	Healthy Pre-Term and Full-Term Neonates	*MIC* = 64 mg/L	Well tolerated
	100 mg/kg every 24 h 2 partial doses	Pre-Term Neonates aged 1–3 days	*MIC* = 32 mg/L	
	150–200 mg/kg every 24 h 3–4 partial doses	Pre-Term neonates aged 3–5 weeks	*MIC* = 32 mg/L	
Fosfomycin [172]	100 mg/kg every 12 h	Neonates aged < 7 days or weighing < 1500 g	*MIC* = 32 mg/L	
Metronidazole [319]	7.5 mg/kg every 12 h 15 mg/kg (LD)	PMA < 34 weeks	*Cmin_ss_ >* 2 mg/L	
	7.5 mg/kg every 8 h 15 mg/kg (LD)	PMA 34–40 weeks		

% fT > *MIC*—Percent of time for free drug concentration remains above the minimum inhibitory concentration; % T > *MIC*—Percent of time for total drug concentration remains above the minimum inhibitory concentration; AUC—Area under curve; BW—Body weight; *C_max_*—Maximum (peak) plasma concentration; *C_min_*—Minimum plasma concentration; *C_ss_*—Steady-state concentration; *C_trough_*—Pre-dose plasma concentration; EOS—Early onset sepsis; f*C_ss_*—free plasma steady—state concentration; GA—Gestational age, weeks; LD—Loading dose; MD—Maintenance dose; MIC—Minimal inhibitory concentration; PCA—Post-conceptual age, weeks; PK/PD—Pharmacokinetics/Pharmacodynamics; PMA—Postmenstrual age, weeks; PNA—Postnatal age, weeks, days, hours.

## Data Availability

Sources of information used in this review are listed in the References.

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
