# Peer review of "Developmental Pharmacokinetics of Antibiotics Used in Neonatal ICU: Focus on Preterm Infants"

_biomedicines, 2023, doi:10.3390/biomedicines11030940_

Round 1

Reviewer 1 Report

Dear Authors,

It is a very well done review, but there are some aspects on which I think some improvements should be made, as follows:

In the section about the absorption of antibiotics, with cephalosporins as an example - lines 263-290 - data from fairly old literature are presented (for example [78] from 1973, [79] 1979, [80] 1978, [81] 1981, [82] 1978, etc.). This class of antibiotics is quite frequently indicated in various infectious pathologies, that's why I suggest the authors to present more recent studies, considering that they are cephalosporins from generations 3-4 and even 5 that are indicated for newborns. Thus, in this section no data are presented at all about cephalosporins from these generations, while in the following sections (distribution, for example) cephalosporins from higher, more recent generations are exemplified, which may be of interest to specialists.

The title of the article is "Developmental pharmacokinetics of antibiotics used in neonatal ICU: focus on preterm infants", so that in lines 378-389 there are data only about adolescents and adults, also in lines 395-399 only about children and adolescents. I suggest the authors to make the connection, or the comparison with newborns, so that it remains within the scope of the article.

The following sections are much more extensive, presenting data from much more recent studies and reserve antibiotics (glycopeptides, lipoglycopeptides, etc.).

Considering that there is also a section related to the dosing of antibiotics, there is no mention of plasma half-life, an extremely important PK parameter for the PK of a drug in establishing both therapeutic doses and the administration interval, especially in context in which data are presented about antibiotics with extremely long T ½ (for example oritavancin -glycopeptide, dalbavancin - lipoglycopeptide, etc.).

In the excretion subsection, for lines 788-794, I suggest the authors to cite data from the literature to support this information.

In conclusion, the authors emphasize the essential importance of antibiotic dosing considering the significant differences between premature newborns and full-term newborns, which leads to the necessity, as we suggested, of inserting some data about T 1/2 in this review.

Best regards

Author Response

  1. In the section about the absorption of antibiotics, with cephalosporins as an example - lines 263-290 - data from fairly old literature are presented (for example [78] from 1973, [79] 1979, [80] 1978, [81] 1981, [82] 1978, etc.). This class of antibiotics is quite frequently indicated in various infectious pathologies, that's why I suggest the authors to present more recent studies, considering that they are cephalosporins from generations 3-4 and even 5 that are indicated for newborns. Thus, in this section no data are presented at all about cephalosporins from these generations, while in the following sections (distribution, for example) cephalosporins from higher, more recent generations are exemplified, which may be of interest to specialists.

Answer: Thank you for comments, in the section Absorption we were aimed on consideration of oral antibiotics absorption, and for Cephalosporines oral forms are available mainly for the 1st and 2nd generations, PK studies of their representatives were performed in 70-s to 90-s period, and no more recent sources are available, unfortunately (search was made using PubMed)

  1. The title of the article is "Developmental pharmacokinetics of antibiotics used in neonatal ICU: focus on preterm infants", so that in lines 378-389 there are data only about adolescents and adults, also in lines 395-399 only about children and adolescents. I suggest the authors to make the connection, or the comparison with newborns, so that it remains within the scope of the article.

Answer: Thank you for the comments, the connection in the text was made.

  1. The following sections are much more extensive, presenting data from much more recent studies and reserve antibiotics (glycopeptides, lipoglycopeptides, etc.). Considering that there is also a section related to the dosing of antibiotics, there is no mention of plasma half-life, an extremely important PK parameter for the PK of a drug in establishing both therapeutic doses and the administration interval, especially in context in which data are presented about antibiotics with extremely long T ½ (for example oritavancin -glycopeptide, dalbavancin - lipoglycopeptide, etc.).

Answer: Thank you for suggestion, we did add information about T1/2 in the Table 9.

  1. In the excretion subsection, for lines 788-794, I suggest the authors to cite data from the literature to support this information.

Answer: Thank you, we did add citing.

  1. In conclusion, the authors emphasize the essential importance of antibiotic dosing considering the significant differences between premature newborns and full-term newborns, which leads to the necessity, as we suggested, of inserting some data about T 1/2 in this review.

Answer: Thank you for suggestion, data on T1/2 (in adults, full-terms, premature ones) was added in the Table 9 for all indicated antibacterials.

Reviewer 2 Report

The submitted review is of readers interest, and it is well structured.

Some small corrections should be  implemented:

1. Figure 1,  oxazolidinones are rather defined as time dependent drugs (not AUC)

2. Grepafloxacin is any longer used in clinical applications, should be deleted

3. cefodezime – name not correct, not existing drug, for correction

4. Instead of MDR1, P-glycoprotein (P-gp) term should be used

5. The expression of OATP1A2 in gastrointestinal tract is questionable, for correction or comments (e.g. Mol Pharm. 2014 Oct 6;11(10):3547-55)

6. Citations of experimental studies should be deleted, e.g. ref. 112 (this information adds no value for clinical applications)

7. Metronidazole (as not only antibiotics, but also chemiotherapeutic agents are discussed in the review) should be included in the review (e.g. ClinicalTrials.gov Identifier: NCT01222585

English should be polished

Author Response

Thank you for your work, we did make corrections according to your comments:

  1. Figure 1, oxazolidinones are rather defined as time dependent drugs (not AUC)

Answer: Fig.1 is changed.

  1. Grepafloxacin is any longer used in clinical applications, should be deleted

Answer: grepafloxacin is deleted from the text.

  1. Cefodezime – name not correct, not existing drug, for correction

Answer: Thank you for the note, we did change the name, correct variant is “cefodizime”

  1. Instead of MDR1, P-glycoprotein (P-gp) term should be used

Answer: MDR1 is replaced in the text and tables on “P-gp”

  1. The expression of OATP1A2 in gastrointestinal tract is questionable, for correction or comments (e.g. Mol Pharm. 2014 Oct 6;11(10):3547-55)

Answer: Thank you for the comments, variability was marked

  1. Citations of experimental studies should be deleted, e.g. ref. 112 (this information adds no value for clinical applications)

Answer: following your advice we did remove sources with experimental data.

  1. Metronidazole (as not only antibiotics, but also chemiotherapeutic agents are discussed in the review) should be included in the review (e.g. ClinicalTrials.gov Identifier: NCT01222585

Answer: Thank you for suggestion, data on metronidazole was added in the Table 9 and Table 10.